# EditInfinity: Image Editing with Binary-Quantized Generative Models

**Jiahuan Wang**[*1] **Yuxin Chen**[*2] **Jun Yu**[1] **Guangming Lu**[†1] **Wenjie Pei**[†1]

[1]Harbin Institute of Technology, Shenzhen  [2]The Hong Kong University of Science and Technology

wangjiahuanszhit@163.com  ychenqa@connect.ust.hk  yujun@hit.edu.cn
luguangm@hit.edu.cn  wenjiecoder@outlook.com

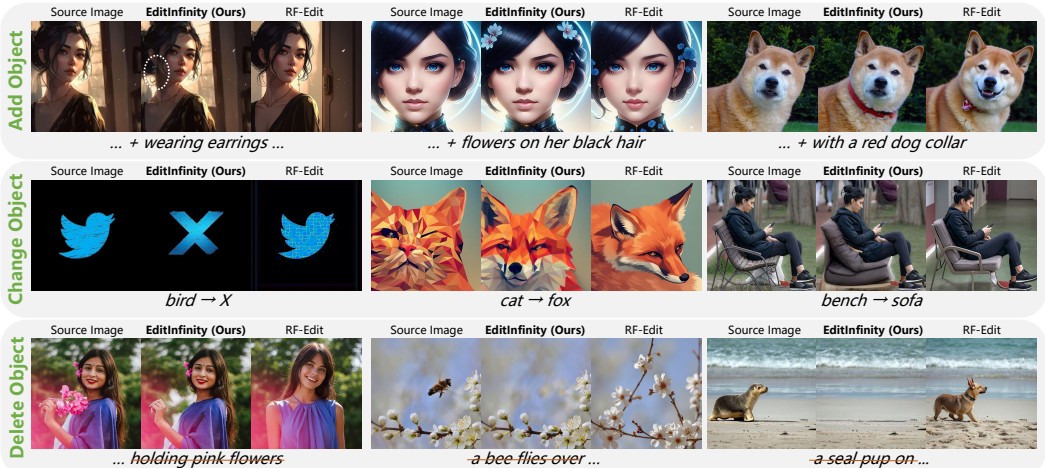

Figure 1: Our method, *EditInfinity*, delivers strong performance in **background preservation** in unedited regions and **text alignment** in edited regions across diverse editing tasks, including add, change and delete object, showing clear advantages over the latest state-of-the-art diffusion-based method RF-Edit [53], as illustrated by representative examples.

## Abstract

Adapting pretrained diffusion-based generative models for text-driven image editing with negligible tuning overhead has demonstrated remarkable potential. A classical adaptation paradigm, as followed by these methods, first infers the generative trajectory inversely for a given source image by image inversion, then performs image editing along the inferred trajectory guided by the target text prompts. However, the performance of image editing is heavily limited by the approximation errors introduced during image inversion by diffusion models, which arise from the absence of exact supervision in the intermediate generative steps. To circumvent this issue, we investigate the parameter-efficient adaptation of binary-quantized generative models for image editing, and leverage their inherent characteristic that the exact intermediate quantized representations of a source image are attainable, enabling more effective supervision for precise image inversion. Specifically, we propose *EditInfinity*, which adapts *Infinity*, a binary-quantized generative model, for image editing. We propose an efficient yet effective image inversion mechanism that integrates text prompting rectification and image style preservation, enabling precise image inversion. Furthermore, we devise a

---

* Equal contribution.  † Corresponding authors.

39th Conference on Neural Information Processing Systems (NeurIPS 2025).

holistic smoothing strategy which allows our *EditInfinity* to perform image editing with high fidelity to source images and precise semantic alignment to the text prompts. Extensive experiments on the PIE-Bench benchmark across 'add', 'change', and 'delete' editing operations, demonstrate the superior performance of our model compared to state-of-the-art diffusion-based baselines. Code available at: `https://github.com/yx-chen-ust/EditInfinity`.

## 1 Introduction

Text-driven image editing aims to modify the content of an image in accordance with the given text prompts while maintaining the integrity of the unedited regions. In contrast to training-from-scratch methods [11, 22, 63] that incur expensive training costs, the adaptation of pre-trained models, particularly diffusion-based generative models, with lightweight fine-tuning overhead has emerged as a predominant paradigm for image editing, demonstrating remarkable potential [29, 21, 43].

A classical adaptation paradigm in diffusion models for image editing [29, 20, 2] consists of two essential steps: 1) image inversion, which aims to infer the generative trajectory along the sampling process in reverse for a given source image, striving to reconstruct the image accurately, and 2) image editing, conducted along the inferred trajectory guided by the target text prompts. Consequently, the precision of image inversion is critical to the performance of image editing. Nevertheless, it is intractable to obtain the exact sampling trajectory of a source image for a pretrained diffusion model. Thus, image inversion is either performed employing the deterministic sampling technique [5, 42, 10, 15, 20, 56] to approximate the intermediate noisy representations along the reversed sampling path, or it is formulated as a optimization problem to finetune the pretrained diffusion model to fit the approximate intermediate results along the sampling path [29, 9]. Consequently, a potential limitation of this adaptation paradigm of diffusion models for image editing is that the performance of image editing is heavily constrained by the approximate errors introduced during image inversion.

To address aforementioned limitation, in this work, we investigate the parameter-efficient adaptation of binary-quantized generative models for image editing. Unlike diffusion models, binary-quantized generative models quantize images into a discrete latent space and model the data distribution in this quantized space for generation. Thus, an inherent characteristic of binary-quantized models is that the exact quantized representations for an arbitrary image can be directly inferred, potentially enabling more precise image inversion. Motivated by this observation, we propose *EditInfinity*, which adapts *Infinity*—a binary-quantized generative model with powerful text-to-image generation capability—for image editing, following the classical 'image inversion-image editing' adaptation paradigm.

Considering a pretrained *Infinity* as a mapping function between the distribution of textual prompts and image data distribution, performing inverse inference on the pretrained model to obtain the exact textual embedding for a source image is intractable, whereas the user-provided source text prompts generally cannot precisely match with the source image. Therefore, we formulate the image inversion process of *EditInfinity* as an optimization problem, aiming to learn an accurate textual embedding for a given source image, guided by provided source text prompts. A notable advantage of this design is that the intermediate multi-scale quantized representations by *Infinity* for the source image can be utilized as exact supervision to optimize the image inversion process, yielding precise image inversion and thereby, high-quality image editing. To conclude, we make the following contribution.

- We propose *EditInfinity*, which apply the classical 'image inversion-image editing' adaptation paradigm to *Infinity*, a prominent binary-quantized model, to investigate the parameter-efficient adaptation of binary-quantized generative models for image editing.

- We design an efficient yet effective image inversion mechanism comprising text prompting rectification and image style preservation, leveraging the quantized representations as exact supervision to enable precise image inversion.

- We devise a holistic smoothing strategy which allows our *EditInfinity* to perform image editing with high fidelity to source image and precise semantic alignment to the text prompts.

- We conduct extensive experiments on the PIE-Bench benchmark and comprehensively demonstrate the superior performance of our *EditInfinity* compared to state-of-the-art diffusion-based approaches across diverse editing operations, excelling in both background preservation and semantic alignment with target text prompts.

## 2 Related Work

### 2.1 Image Editing with Diffusion Models

Image editing researches [27, 16, 21, 31, 4] have been predominantly driven by diffusion models [36, 38, 33, 22], and are broadly categorized into training-based and training-free paradigms [40]. Training-based methods [3, 58, 11, 39, 18, 24, 22] achieve impressive editing capabilities, but their requirement for an expensive training dataset limits practical applicability. In contrast, training-free [48, 23, 43] methods have emerged as a more flexible alternative, establishing an inversion-editing paradigm [40]. The inversion stage focuses on accurate latent code inversion. Recent works [29, 9, 28, 20, 53] have developed improved inversion samplers to ease the inherent reconstruction inaccuracies. In the editing stage, numerous methods [16, 47, 4, 1, 32, 25, 26] leverage attention in diffusion models to edit while preserving overall image structure. Despite these advances, a key limitation remains: existing methods fail to preserve both text alignment fidelity and source image consistency. This trade-off between editability and faithfulness motivates us to investigate more robust editing frameworks.

### 2.2 Autoregressive Image Generation Models

Autoregressive models have demonstrated remarkable scalability in image generation by leveraging next-token prediction, a paradigm inherited from LLMs (**L**arge **L**anguage **M**odels) [52]. Early methods like PixelCNN [49] and PixelRNN [50] model pixels directly, but their quadratic dependency growth makes high-resolution generation impractical. Thus, subsequent works avoid modeling the data distribution directly in pixel space and instead model it in a compact latent space. As a pioneering work, VQVAE [51] constructs a discrete latent space by vector quantization and learns the underlying latent distribution by autoregressive models. Recently, VAR (**V**isual **A**uto**R**egressive Modeling) [46] reformulates autoregressive image generation as a next-scale prediction task, capturing global structural priors to achieve state-of-the-art generation quality while improving sampling speed.

The success of autoregressive models naturally extends to text-to-image generation. Pioneering works like DALL-E [35] and CogView [8] unify text and image tokens within a single transformer decoder. Subsequently, Parti [57] and LlamaGen [44] decouple text and image processing by employing dedicated text encoders to guide the autoregressive decoder. Then, HART [45] integrates VAR's hybrid tokenizers to improve generation quality. Latest, Infinity [14] advances autoregressive image generation by introducing Bitwise Visual AutoRegressive Modeling. It establishes a new foundational model for autoregressive text-to-image models and achieves competitive results with diffusion-based approaches. As our method builds on Infinity, we outline its architecture in Section 3.

## 3 Preliminary: Infinity

**Bitwise Multi-scale Residual Quantization.** An image $I \in \mathbb{R}^{H \times W \times 3}$ is first encoded into the original feature $F$, which is then tokenized into bitwise multi-scale residual maps $\{R_k\}_{k=1}^{K}$ through iterative residual approximation. At scale $k$, residual features are computed between the original feature $F$ and the cumulative feature $F_{k-1}$ from previous scales.

$$z_k = \text{down}\left(F - F_{k-1}, (h_k, w_k)\right) \in \mathbb{R}^{h_k \times w_k \times d}, \tag{1}$$

where $\text{down}(\cdot)$ performs bilinear downsampling to target resolution $(h_k, w_k)$. To quantize residuals, Infinity adopts BSQ (**B**inary **S**pherical **Q**uantization) [61]:

$$R_k = \mathcal{Q}(z_k) = \frac{1}{\sqrt{d}}\text{sign}\left(\frac{z_k}{\|z_k\|}\right) \in \{\frac{-1}{\sqrt{d}}, \frac{1}{\sqrt{d}}\}^{h_k \times w_k \times d}. \tag{2}$$

Then, the cumulative feature $F_k$ at scale $k$ is computed recursively:

$$F_k = \sum_{i=1}^{k} \text{up}(R_i, (h_K, w_K)) \in \mathbb{R}^{h_K \times w_K \times d}, \tag{3}$$

where $\text{up}(\cdot)$ denotes bilinear upsampling.

**Bitwise Autoregressive Modeling.** The transformer predicts residuals autoregressively across $K$ scales, conditioned on the prompt $t$. Formally, the autoregressive likelihood is:

$$p(R_{1:K}|\Psi(t)) = \prod_{k=1}^{K} p\big(R_k | \underbrace{R_1, \ldots, R_{k-1}}_{\text{all previous scales}}, \Psi(t)\big),\tag{4}$$

where $\Psi(\cdot)$ denotes Flan-T5 [6]. To tackle the large codebook challenge, Infinity proposes the Infinite-Vocabulary Classifier, which decomposes the prediction into $d$ independent binary classifiers.

## 4 Method

Successful image editing requires precise content modifications that semantically align with target prompts while remaining faithful to unedited regions. To this end, a classical adaptation paradigm repurposes a pretrained text-to-image generative model to image editing through two steps: 1) image inversion, which inversely infers the corresponding generative trajectory for the source image by reversing the sampling process, and 2) image editing, performed along the inferred generative trajectory guided by the target text prompts. Our proposed *EditInfinity* applies this paradigm to Infinity [14], a binary-quantized generative model, harnessing the inherent characteristics of quantized generative models to potentially achieve precise image inversion and high-quality image editing.

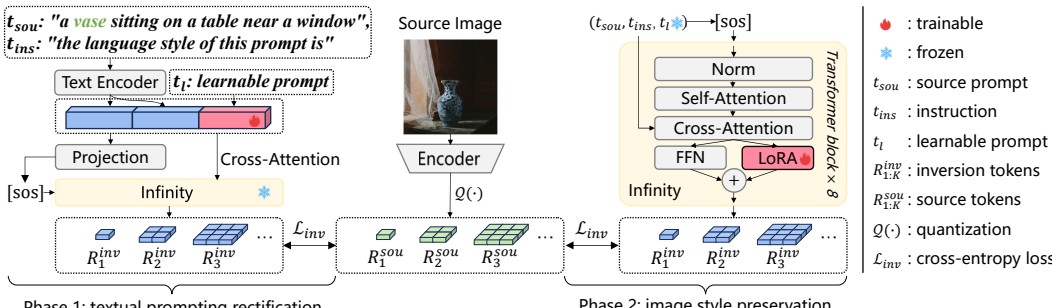

Figure 2: **Image Inversion with Exact Supervision.** Given a source image $I_{sou}$ and its prompt $t_{sou}$, we first quantize $I_{sou}$ into exact tokens $R_{1\ldots K}^{sou}$. Then, we concatenate $t_{sou}$ with an instruction $t_{ins}$ and a learnable prompt $t_l$, which is optimized via $\mathcal{L}_{inv}$ under the supervision of $R_{1\ldots K}^{sou}$. Afterwards, the prompt is frozen, and LoRA is applied to the FFN layers of Infinity to further reconstruct $I_{sou}$.

### 4.1 Image Inversion with Exact Supervision

A text-to-image generative model performs image generation by learning a mapping from the distribution of text prompts to image data distribution. However, since the mapping function is unknown, it is intractable to inversely obtain the exact textual embedding for a given image. Meanwhile, the user-provided source text prompt generally cannot precisely match the source image. To circumvent this problem, we formulate the image inversion process as an optimization problem with exact supervision to infer the text embedding precisely matched with the source image:

$$\mathcal{L}_{inv} = -\frac{1}{K} \sum_{k=1}^{K} \big(R_k^{sou} \cdot \log p(R_k^{inv}|R_{<k}^{sou}, \Psi(t))\big),\tag{5}$$

where $\mathcal{L}_{inv}$ is formulated as a cross-entropy loss applied to each inversion token $R_k^{inv}$. Compared to the diffusion-based models for image editing, a key advantage of binary-quantized generative models is that the exact groundtruth of the intermediate outputs ($R_{1\ldots K}^{sou}$ in 'Infinity') for a given image along the generative trajectory is attainable by token-wise quantization, enabling exact supervision for optimization of image inversion in Equation 5.

**Textual prompting rectification.** To guide the optimization of Equation 5 toward a text embedding that matches the source image, we treat the source prompt $t_{sou}$ as a reference and apply text prompting tuning to rectify it into a semantically aligned textual condition. Concretely, we first augment $t_{sou}$

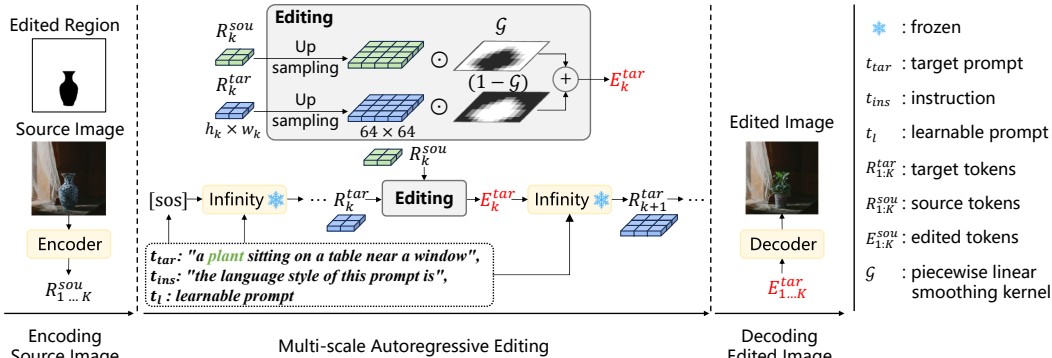

Figure 3: **Image Editing with Holistic Smoothing.** First, source image is encoded into $R^{sou}_{1\dots K}$. At each step $k$ of autoregressive generation, generated $R^{tar}_k$ is conditioned on the concatenation of the target prompt $t_{tar}$, instruction $t_{ins}$, and optimized learnable prompt $t_l$ and then, is blended with $R^{sou}_k$ guided by piecewise linear smoothing kernel $\mathcal{G}$, forming edited tokens $E^{tar}_k$ to prepare for guiding the next-scale generation. Finally, $E^{tar}_{1\dots K}$ is decoded into the edited image.

with 20 learnable prompt tokens $t_l$ and an instruction prompt $t_{ins}$ (e.g., "the language style of this prompt is") to bridge the semantic gap between the source prompts and the solution. Second, we pass $t_{sou}$ and $t_{ins}$ through the text encoder $\Psi(\cdot)$ of Infinity to obtain text embeddings $\Psi(t_{sou}, t_{ins})$. We then concatenate those embeddings with $t_l$ to form the textual conditioning input $[\Psi(t_{sou}, t_{ins}), t_l]$ for Infinity. Finally, we freeze all Infinity parameters and optimize only $t_l$ using cross-entropy loss, where the supervision signals are exact tokens $R^{sou}_{1\dots K}$ derived from the source image.

**Image style preservation.** While the learnable prompt adapt the semantic content, they may fall short in preserving structural style characteristics. The low-rank bias [17, 19, 41] (rank $\ll \dim(W)$) favors smooth and global modifications to the output distribution, thereby encouraging reconstructions that preserve overall structure and appearance while avoiding overfitting to high-frequency artifacts. To this end, we employ LoRA [17] to refine the pretrained weights $W$ with minimal overhead $\Delta W$ (only inserts trainable low-rank matrices into FFN layers [52]) after rectifying textual prompt. Then, the learned $\Delta W$ is retained during editing, allowing the model to faithfully preserve global style traits of the source image even when applying novel target prompts.

## 4.2 Image Editing with Holistic Smoothing

We aim to manipulate only the desired regions while preserving the structural integrity of unedited areas. To this end, we introduce a precise token replacement strategy that enables localized, semantically aligned editing at the token level. Given the optimized learnable prompt $t_l$ and LoRA $\Delta W$, we perform conditional generation under the target prompt $t_{tar}$, instruction $t_{ins}$ and optimized learnable prompt $t_l$, which ensures the edited image adheres to the target semantics while maintaining structural fidelity with the source image.

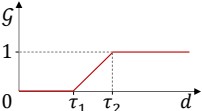

Figure 4: $\mathcal{G}$ as a function of $d$, enabling smooth transitions from edited to unedited regions.

**Piecewise Linear Smoothing Kernel.** The core idea of our editing paradigm is to construct the edited tokens $E^{tar}_{1:K}$ by blending source tokens $R^{sou}_{1:K}$ and target tokens $R^{tar}_{1:K}$ in a spatially controlled manner. A direct blend will result in a splicing phenomenon, so we first localize the edit with a user-provided mask $M$—a standard setting in image editing [30, 62] where text-only prompts often lack spatial specificity [16, 37, 59]. Then, we define a piecewise linear smoothing kernel $\mathcal{G}$ to guide the blending. Specifically, $\mathcal{G}$ is defined over the Manhattan distance $d$ to calculate location weights per location, as in Equation 6:

$$\mathcal{G}^{i,j} = \begin{cases} 0, & d^{i,j} \leq \tau_1 \\ \frac{d^{i,j} - \tau_1}{\tau_2 - \tau_1}, & \tau_1 < d^{i,j} < \tau_2 \ , \\ 1, & d^{i,j} \geq \tau_2 \end{cases} \quad d^{i,j} = \min_{(x,y) \in M} \left( |i - x| + |j - y| \right) . \tag{6}$$

Here, $d^{i,j}$ denotes the Manhattan distance from token $(i, j)$ to the nearest token within $M$. The kernel $\mathcal{G}^{i,j}$ is designed to gradually transition from 0 to 1 within a controllable band defined by thresholds $\tau_1$ and $\tau_2$, which is visualized in Figure 4. Specifically, tokens within a distance of $\tau_1$ from the edit region are assigned zero weight to encourage full preservation from target content, while those beyond $\tau_2$ are fully replaced by the source. Tokens in the intermediate band are assigned weights via linear interpolation, facilitating smooth blending between source and target content. This formulation effectively suppresses boundary artifacts by promoting seamless transitions between source and edited regions.

**Multi-scale Autoregressive Editing.** Building on image inversion and the piecewise linear smoothing kernel $\mathcal{G}$, we realize image editing as a multi-scale autoregressive token-replacement process. At each scale, generated target tokens are blended with source tokens under spatial weights provided by $\mathcal{G}$, and the resulting edited tokens serve as context for the next scale. This coarse-to-fine schedule localizes semantic changes to the masked region while preserving global structure elsewhere. Algorithm 1 details the procedure.

Our algorithm begins by quantizing $I_{sou}$ to extract precise source tokens $R_{1:K}^{sou}$. At each scale $k$, Infinity$(\cdot)$ generates the target token $R_k^{tar}$ conditioned on previous tokens $\hat{R}_{<k}^{tar}$ and prompts embedding $[\Psi(t_{tar}, t_{ins}), t_l]$. We then upsample $R_k^{tar}$ and $R_k^{sou}$ to $(h_K, w_K)$ and blend them under the guidance of $\mathcal{G}$ to obtain the edited token $E_k^{\text{tar}}$. This aligns edited regions with target semantics while preserving source fidelity elsewhere. If $k < K$, we downsample $E_k^{\text{tar}}$ to $(h_{k+1}, w_{k+1})$ to form $\hat{R}_k^{\text{tar}}$, which serves as the autoregressive state at the next scale, allowing blended semantics and structure to propagate across scales. After traversing all scales, edited tokens $E_{1:K}^{tar}$ are decoded to edited image $I_{tar}$.

---

**Algorithm 1** Multi-scale Autoregressive Editing

1: **Inputs:** source image $I_{sou}$; target prompt $t_{tar}$; instruction $t_{ins}$; optimized learnable prompt $t_l$;
2: **Hyperparameters:** scales $K$, resolutions $(h_k, w_k)_{k=1}^K$
3: $R_{1...K}^{sou} = \mathcal{Q}(\mathcal{E}(I_{sou}))$ ▷ $\mathcal{E}$: encoder; $\mathcal{Q}$: quantizer
4: $[\Psi(t_{tar}, t_{ins}), t_l]$ projected into $\hat{R}_0^{tar}$ (i.e., [sos])
5: **for** $k = 1$ **to** $K$ **do**
6:    $R_k^{tar} = \text{Infinity}(\hat{R}_{<k}^{tar}, [\Psi(t_{tar}, t_{ins}), t_l])$
7:    $E_k^{tar} = \text{Upsample}(R_k^{tar}, (h_K, w_K)) \odot (1 - \mathcal{G}) + \text{Upsample}(R_k^{sou}, (h_K, w_K)) \odot \mathcal{G}$
8:    **if** $k < K$ **then**
9:        $\hat{R}_k^{tar} = \text{Downsample}(E_k^{tar}, (h_{k+1}, w_{k+1}))$
10:   **end if**
11: **end for**
12: $I_{tar} = \mathcal{D}(E_{1...K}^{tar})$ ▷ $\mathcal{D}$: decoder
13: **Return** edited Image $I_{tar}$

---

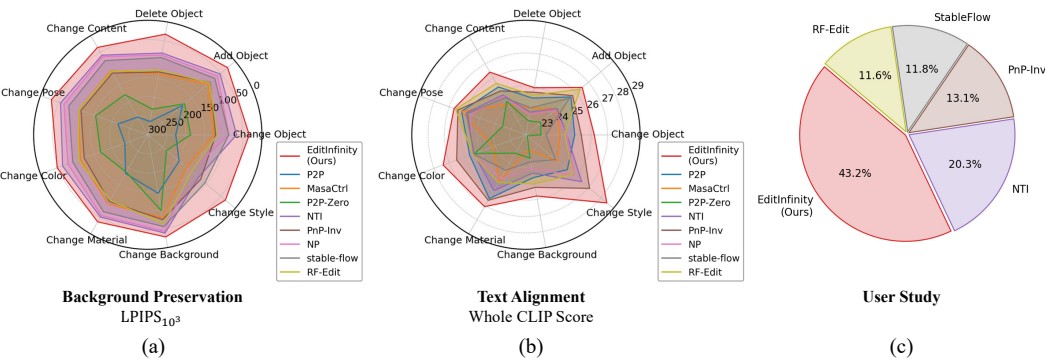

Figure 5: **Comprehensive performance evaluation on PIE-Bench.** (a) and (b) report background preservation and text alignment metrics across nine tasks. (c) summarizes user study preferences.

# 5 Experiments

## 5.1 Experimental Setup

**Comparison Methods.** We compare our method against a range of methods. (1) Open-source methods: including Diffusion UNet models—P2P [16], MasaCtrl [4], P2P-Zero [31], NTI [29], PnP-Inv [20], and NP [28]—and Diffusion Transformer models—StableFlow [2] and RF-Edit [53]. (2) Closed-source method: Gemini 2.0 [13], a current frontier of large-scale commercial model.

**Benchmark.** We conduct comprehensive experiments on PIE-Bench (**P**rompt-based **I**mage **E**diting **Bench**mark) [20], the prevailing standard in image editing evaluation. This benchmark contains 700 test cases covering nine editing types. Each case provides a source image with a corresponding prompt, target editing prompt, and the editing mask.

**Metrics.** Our evaluation employs seven carefully selected metrics across two critical dimensions. For background preservation, we use four complementary metrics: PSNR and MSE for pixel-level accuracy, LPIPS [60] for perceptual similarity, and SSIM [54] for structural similarity. For text-image alignment, we report CLIP scores [34] of the whole image and the edited region with the target prompt. Additionally, we adopt IR (**I**mage **R**eward [55]), a learned metric trained on human preference data, specifically sensitive to editing failures, often assigning negative scores to failed outputs.

**Implementation Details.** We implement our method based on Infinity-2B [0]. For editing, we set $\tau_1 = 1$ and $\tau_2 = 4$ in Equation 6. Inversion is trained on two NVIDIA L20 GPUs, and editing runs on a single NVIDIA L20 GPU. Refer to Supplementary Material A.2 for more details.

Table 1: **Quantitative results on PIE-Bench.** In the 'Base Model' column, 'U', 'T', and 'A' represent Diffusion UNet, Diffusion Transformer, and Autoregressive models, respectively. Diffusion UNet models employ Stable Diffusion v1.4, with the exception of PnP-Inv, which utilizes v1.5. Diffusion Transformer models leverage FLUX.1-dev, while Autoregressive models use Infinity.

| Method | Venue | Base Model | Background Preservation | | | | Text Alignment | | |
|---|---|---|---|---|---|---|---|---|---|
| | | | PSNR↑ | LPIPS$_{10^3}$↓ | MSE$_{10^4}$↓ | SSIM$_{10^2}$↑ | Whole↑ | Edited↑ | IR$_{10}$↑ |
| P2P[16] | ICLR'23 | | 17.87 | 208.80 | 219.88 | 71.14 | 25.01 | 22.44 | 0.017 |
| MasaCtrl[4] | ICCV'23 | | 22.17 | 106.62 | 86.97 | 79.67 | 23.96 | 21.16 | -1.66 |
| P2P-Zero[31] | SIGGRAPH'23 | | 20.44 | 172.22 | 144.12 | 74.67 | 22.80 | 20.54 | -6.59 |
| NTI[29] | CVPR'23 | U | 27.03 | 60.67 | 35.86 | 84.11 | 24.75 | 21.86 | 2.77 |
| PnP-Inv[20] | ICLR'24 | | 22.46 | 106.06 | 80.45 | 79.68 | 25.41 | 22.62 | 4.17 |
| NP[28] | WACV'25 | | 26.21 | 69.01 | 39.73 | 83.40 | 24.61 | 21.87 | 2.42 |
| StableFlow[2] | CVPR'25 | T | 21.64 | 92.28 | 115.21 | 84.94 | 24.65 | 21.70 | 1.88 |
| RF-Edit[53] | ICML'25 | | 23.22 | 131.18 | 75.00 | 81.44 | 25.22 | 22.40 | 5.18 |
| Gemini[13] | - | - | 23.22 | 105.17 | 188.63 | 81.10 | 25.28 | 22.28 | 5.30 |
| **EditInfinity** | NeurIPS'25 | A | **27.95** | **33.08** | **24.27** | **92.12** | **26.41** | **23.47** | **5.88** |

Table 2: **Evaluation of Base Models on the GenEval Benchmark.** When evaluating Infinity, we adopt the same evaluation protocol as used for Stable Diffusion v1.4, v1.5, and FLUX.1-dev, i.e., without prompt rewriting.

| Base Model | Overall | Single Object | Two Object | Counting | Colors | Position | Attribute Binding |
|---|---|---|---|---|---|---|---|
| Stable Diffusion v1.4 | 0.42 | 0.97 | 0.39 | 0.33 | 0.73 | 0.03 | 0.05 |
| Stable Diffusion v1.5 | 0.43 | 0.97 | 0.38 | 0.35 | 0.76 | 0.04 | 0.06 |
| **FLUX.1-dev** | **0.66** | **0.98** | **0.81** | **0.74** | 0.79 | 0.22 | 0.45 |
| **Infinity-2B** | **0.66** | **0.98** | 0.78 | 0.63 | **0.83** | **0.25** | **0.53** |

## 5.2 Comparison to State-of-the-Arts

**Quantitative Results.** As demonstrated in Table 1, our method sets a new state-of-the-art in text-driven image editing by significantly improving the trade-off between two key objectives: (1) rigorous background preservation and (2) precise text-aligned editing. While existing methods struggle with this inherent trade-off, our framework achieves a superior balance, outperforming all others by notable margins in both aspects. Notably, our method attains the best IR$_{10}$ score (5.88), reflecting substantially higher editing success rates than competing methods. We further provide task-wise comparisons of LPIPS and full CLIP scores across all edit types. As shown in Figure 5 (a) and (b), these results consistently validate the effectiveness of our method across diverse editing scenarios.

---

[0] https://huggingface.co/FoundationVision/Infinity/blob/main/infinity_2b_reg.pth

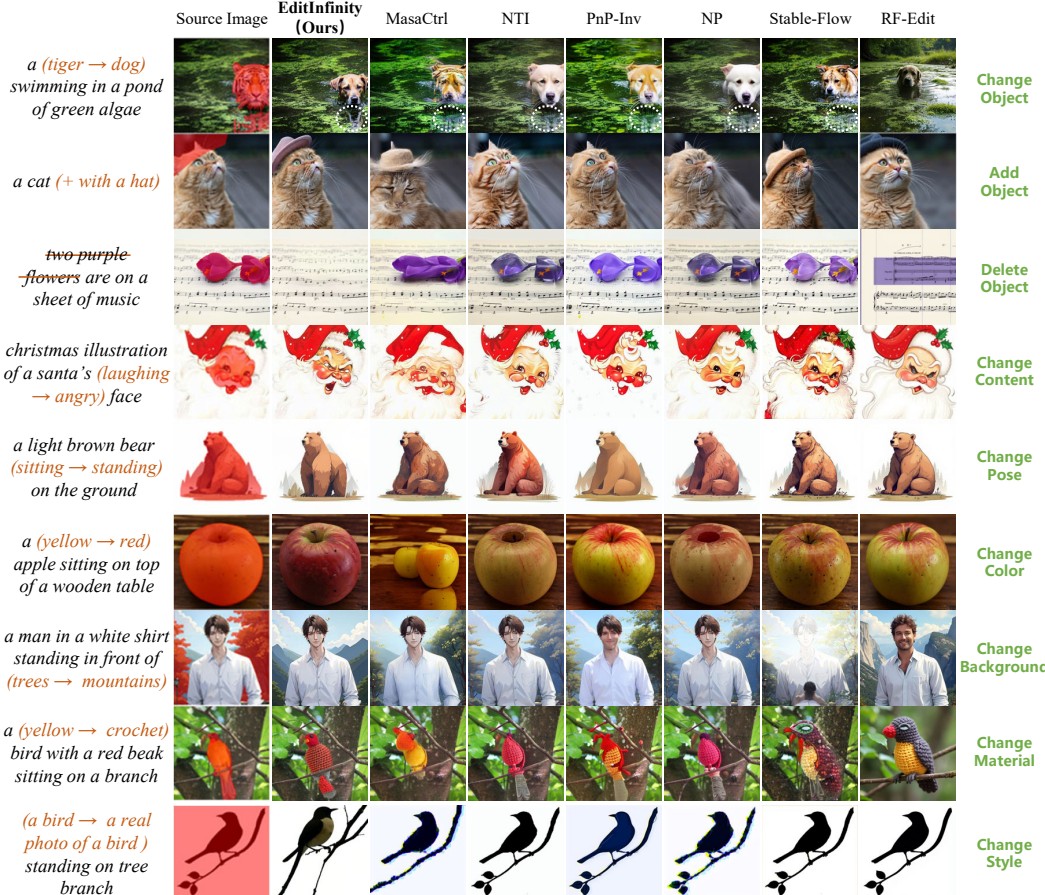

Figure 6: **Qualitative results on PIE-Bench across all nine tasks.** The red mask denotes the edited region $M$, expected to follow the target prompt, while other regions retain the background.

To ensure that the advantage of our method is not only attributed to the stronger generative capacity of base model, we further analyze the base models used by all compared methods. Although our framework is lightweight and tailored to Infinity, its reliance on the base model's generative capacity is consistent with other editing paradigms. As reported in Table 2, we evaluate each method's base model on the GenEval benchmark [12]. Infinity performs comparably to the popular FLUX [22] and even underperforms in certain tasks (e.g., two-object and counting). Nevertheless, our Infinity-based approach surpasses FLUX-based methods such as StableFlow and RF-Edit by a large margin, demonstrating the effectiveness of our method despite the base model not having a clear advantage.

**Qualitative Results.** Visual quality is critical for evaluating image editing. Figure 6 presents qualitative comparisons across all PIE-Bench tasks. For space, we omit P2P and P2P-Zero, which show weaker background preservation and text alignment, respectively (see Table 1). Our method achieves a better trade-off between preserving unedited regions and accurately aligning edited regions with the target prompt. More visualizations are provided in Supplementary Material A.5.

**User Study.** Our method compares against two UNet-based and two transformer-based diffusion models, all showing competitive performance in Table 1. The study uses 140 images from the 'random class' in PIE-Bench [20], covering all editing types. Each of the 60 volunteers is randomly assigned 20 editing cases. For each case, they are shown a source image, a target text prompt, and five edited results (randomly ordered and anonymized). Volunteers selected the best result via a custom web interface, as shown in Supplementary Material Figure 10. Results in Supplementary Material Figure 5 (c) show $43.2\%$ preferred our method—the highest among all approaches, confirming that it maintains strong subjective visual quality.

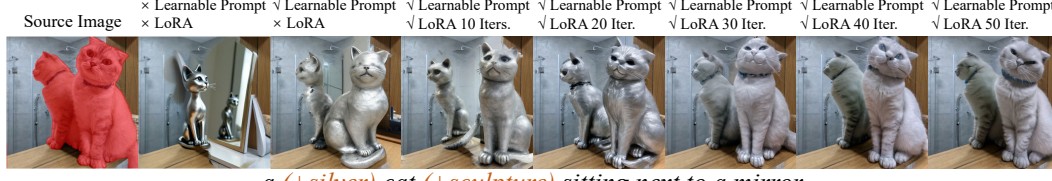

| Source Image | × Learnable Prompt × LoRA | √ Learnable Prompt × LoRA | √ Learnable Prompt √ LoRA 10 Iters. | √ Learnable Prompt √ LoRA 20 Iter. | √ Learnable Prompt √ LoRA 30 Iter. | √ Learnable Prompt √ LoRA 40 Iter. | √ Learnable Prompt √ LoRA 50 Iter. |

*a (+silver) cat (+sculpture) sitting next to a mirror*

Figure 7: Illustrations of ablating the Learnable Prompt and LoRA.

**Runtime Comparison.** We conduct a runtime comparison of our method and other methods on a single NVIDIA L20 GPU, measuring both inversion and editing time, as shown in Table 3. A key advantage of our method is efficient support for multiple edits on the same image, a common real-world scenario. Once the inversion for a given image is completed, subsequent edits can be performed within 3.64 seconds, which is over $7\times$ faster than other methods on average, while the initial inversion time is only $4\times$ longer than other methods on average. This design effectively front-loads the computational cost, making it ideal for iterative workflows.

Table 3: **Runtime comparison.** Time for $n$ edits on an image equals Inversion $+ n \times$ Per-editing.

| Method | Inversion (s) | Per-editing (s) |
|---|---|---|
| P2P[16] | 14.40 | 10.28 |
| **MasaCtrl[4]** | **5.19** | 17.45 |
| P2P-Zero[31] | 13.31 | 62.29 |
| NTI[29] | 95.54 | 10.32 |
| PnP-Inv[20] | 8.32 | 9.54 |
| NP[28] | 9.00 | 10.37 |
| StableFlow[2] | 13.85 | 27.20 |
| RF-Edit[53] | 55.48 | 54.07 |
| **EditInfinity** | 107.06 | **3.64** |

## 5.3 Ablation Study

Ablation studies are performed on the 'random class' of PIE-Bench, covering all types of editing and allowing an efficient and unbiased evaluation.

**Ablation on Learnable Prompt and LoRA.** We design a precise image inversion by leveraging quantized tokens as exact supervision. It integrates the learnable prompt for textual rectification and a LoRA for style preservation. As shown in Figure 7, removing both components causes significant structural inconsistencies. The learnable prompt improves alignment with the target prompt but often shifts global style. Adding LoRA further restores stylistic consistency with the source image. However, prolonged training leads to overfitting, causing the model to ignore editing intents. To balance editability and fidelity, we stop training LoRA after 20 iterations, as shown in Figure 8.

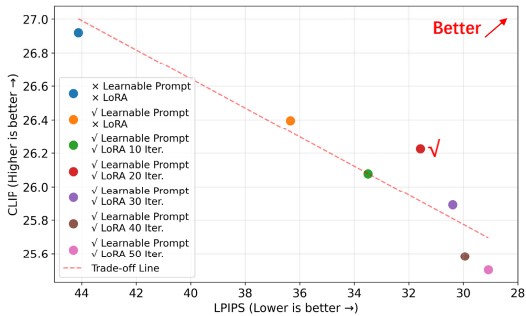

Figure 8: Quantitative results of ablating the Learnable Prompt and LoRA.

**Ablation on Piecewise Linear Smoothing Kernel.** We introduce $\mathcal{G}$ to ensure smooth transitions between edited and unedited regions and to suppress boundary artifacts. As shown in Figure 9 (c), removing $\mathcal{G}$ results in sharp discontinuities along object boundaries (e.g., the cat's ears), confirming its effectiveness in producing seamless edits. To further examine the choice of smoothing function, we compare the linear kernel defined in Equation 6 with a Gaussian kernel ($1 - e^{-d^2/2\alpha^2}$). With proper hyperparameter tuning, the linear kernel achieves superior results, as reported in Table 4. Complete results under both settings are provided in Supplementary Material (Tables 9 and 10).

**Ablation on Mask.** While our method defaults to user-provided masks, it can also leverage Infinity's cross-attention maps [16] for automatic mask generation without modifying the framework. Specifically, we automatically align differing words $x$ between the source and target prompts. After completing inversion, we input the source or target prompt containing $x$ into $\mathrm{Infinity}(\cdot)$ and extract the cross-attention map corresponding to $x$. A threshold is then applied: values above the threshold are set to 0 (mask foreground), and others to 1 (background). Table 5 shows that our method

Table 4: Quantitative results of ablating the Piecewise Linear Smoothing Kernel.

| $\mathcal{G}$ | Background Preservation | | | | Text Alignment | | |
|---|---|---|---|---|---|---|---|
| | PSNR↑ | LPIPS$_{10^3}$↓ | MSE$_{10^4}$↓ | SSIM$_{10^2}$↑ | Whole↑ | Edited↑ | IR$_{10}$↑ |
| ✗ | **31.12** | **24.47** | **13.03** | **93.53** | 25.44 | 23.12 | 2.85 |
| Gaussian kernel | 28.15 | 32.91 | 24.40 | 92.17 | 26.10 | 23.81 | 4.61 |
| Linear kernel | 28.50 | 31.58 | 22.94 | 92.36 | **26.22** | **23.99** | **5.39** |

Table 5: **Quantitative results of ablating the mask.** EditInfinity-u denotes user-provided masks, while EditInfinity-c denotes masks automatically generated via cross-attention. Best and second-best results are shown in **bold** and *italics*, respectively.

| Method | Base Model | Background Preservation | | | | Text Alignment | | |
|---|---|---|---|---|---|---|---|---|
| | | PSNR↑ | LPIPS$_{10^3}$↓ | MSE$_{10^4}$↓ | SSIM$_{10^2}$↑ | Whole↑ | Edited↑ | IR$_{10}$↑ |
| NTI[29] | U | *28.08* | 57.94 | 36.10 | 85.17 | 24.71 | 22.51 | 3.63 |
| RF-Edit[53] | T | 27.26 | 92.27 | *34.46* | 86.67 | 24.65 | 22.03 | 0.61 |
| *EditInfinity-c* | A | 27.47 | *44.97* | 46.91 | *90.30* | *25.71* | *23.22* | **5.40** |
| **EditInfinity-u** | A | **28.50** | **31.58** | **22.94** | **92.36** | **26.22** | **23.99** | *5.39* |

is not highly sensitive to the source of the mask—strong performance is achieved in both cases. Comprehensive comparisons are reported in Supplementary Material Table 11.

**Ablation on Multi-scale Autoregressive Editing.** By autoregressively blending source and target tokens (AR), source tokens in un-edited regions effectively guide the generation of editing regions. As illustrated in Figure 9 (b), blending source tokens at the end of autoregressive generation (NAR, Non-Autoregressive) results in incoherent and visually inconsistent edits. Thus, incorporating guidance at every scale is essential for producing harmonious and realistic results. The quantitative comparison is represented in Supplementary Material Table 12.

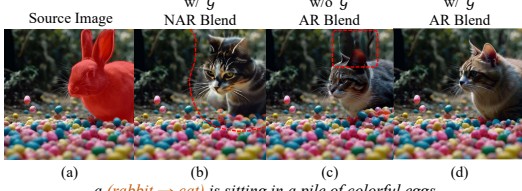

a (*rabbit → cat*) is sitting in a pile of colorful eggs

Figure 9: Illustrations of ablating the Piecewise Linear Smoothing Kernel and Multi-scale Autoregressive Editing.

## 6 Conclusion

We present *EditInfinity*, a parameter-efficient adaptation of binary-quantized generative models for text-driven image editing, following the classical 'inversion–editing' paradigm. During inversion, we formulate the process as an optimization problem supervised by the exact intermediate quantized representations. During editing, we propose a holistic smoothing strategy to blend source and target tokens, preserving unedited regions while aligning with target prompts. Experiments on PIE-Bench show that *EditInfinity* outperforms diffusion-based baselines.

## 7 Limitation

While our method demonstrates strong performance across diverse editing tasks, it shows limitations in extreme cases such as style change, where no background needs to be preserved, and the image contains detailed structural patterns. In such cases, the blending between source and target tokens is constrained, which may lead to suboptimal preservation of structural fidelity from the original image. Nonetheless, thanks to our image inversion strategy, which effectively learns the generative trajectory of source image, our method can still accomplish intended edits, despite slight structural degradation. In contrast, other methods often fail in such challenging scenarios. For example, as shown in Figure 6, row 9, while other methods are unable to convert the painted bird into a realistic one, our method successfully achieves the style change, with only minor deviation in the bird's head pose.

# 8 Acknowledgements

This work was supported in part by the National Natural Science Foundation of China (Grant NO. 62572145, 62176077, 62372133, 62125201 and U24B20174), in part by the Shenzhen Key Technical Project (Grant NO. JCYJ20241202123728037, JSGG20220831092805009, JSGG20220831105603006 and KJZD20230923115117033), in part by the Guangdong Provincial Key Laboratory of Novel Security Intelligence Technologies (Grant NO. 2022B1212010005) and in part by "Guangdong Special Support Plan" (Grant No. 2023TQ07A784).

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

# A  Supplementary Material

The supplementary material is organized as follows:

- Subsection A.1 presents two applications of our method: facial attribute editing and complex-scene image editing.

- Subsection A.2 shows the supplementary implementation details of EditInfinity.

- Subsection A.3 presents more details of user study.

- Subsection A.4 provides more comprehensive ablation studies.

- Subsection A.5 exhibits additional qualitative results for supplementary.

- Subsection A.6 declares broader impacts of our proposed EditInfinity.

- Subsection A.7 declares safeguards of our proposed EditInfinity.

- Subsection A.8 states ethical considerations for EditInfinity.

## A.1  Applications.

**Facial Attribute Change.** To verify that our method generalizes to *unmaskable* edits, where localized masks are impractical, we conduct experiments on facial attribute modification. Specifically, we randomly select 20 images from FFHQ to perform unmasked edits, including age, expression, skin tone. Since the setting is unmasked, there is no background to preserve, and thus standard metrics that rely on background consistency are not applicable. In addition to retaining the Whole metric (CLIP score between the entire edited image and the target prompt), we introduce ArcFace [7] for evaluating identity preservation, and CLIP-I for measuring similarity between the source and edited images. Table 6 shows that our method outperforms strong baselines, including the leading Diffusion UNet model NTI [29] and the Diffusion Transformer model RF-Edit [53]. Thanks to our proposed image inversion algorithm, which effectively learns the generative trajectory of the source image, our method can accomplish the intended edits.

Table 6: Quantitative results on facial images from FFHQ.

| Method | Base Model | ArcFace↑ | CLIP-I↑ | Whole↑ |
|---|---|---|---|---|
| NTI[29] | U | 0.56 | 0.83 | 23.67 |
| RF-Edit[53] | T | 0.61 | 0.79 | 23.54 |
| **EditInfinity** | A | **0.63** | **0.86** | **24.82** |

**Complex Scene Images Editing.** Given that PIE-Bench already contains nearly 50% natural images, it serves as a comprehensive benchmark for evaluating our method on open-ended editing tasks. However, to further assess performance on *complex scenes* involving multiple interacting objects, we conduct an additional evaluation on complex scene images editing. We select 20 MagicBrush images (due to time constraints) filtered by GPT-4o, comprising five samples each with 2, 3, 4, and 5 primary objects. Table 7 demonstrates the superiority of our method in handling complex scenes, compared to the two strong baselines, i.e., NTI [29] and RF-Edit [53].

Table 7: Quantitative results on the complex scene images from MagicBrush.

| Method | Base Model | Background Preservation | | | | Text Alignment | |
|---|---|---|---|---|---|---|---|
| | | PSNR↑ | LPIPS$_{10^3}$↓ | MSE$_{10^4}$↓ | SSIM$_{10^2}$↑ | Whole↑ | Edited↑ |
| NTI[29] | U | 8.81 | 452.03 | 1380.48 | 39.63 | 19.90 | 16.83 |
| RF-Edit[53] | T | 26.00 | 121.84 | 33.98 | 84.73 | 24.13 | 18.29 |
| **EditInfinity** | A | **31.23** | **24.30** | **9.90** | **91.70** | **24.19** | **20.07** |

## A.2 Supplementary Implementation Details.

During image inversion, we set the learning rate to 4.6875e-5 and use AdamW optimizer ($\beta_1 = 0.9$, $\beta_2 = 0.97$) for both the learnable prompt and LoRA training. The two components are optimized sequentially, starting with the learnable prompt, followed by LoRA. To accelerate the convergence of training LoRA, a KL-divergence loss is introduced in addition to the standard cross-entropy loss. Typically, the learnable prompt is trained for 10 iterations, while LoRA is trained for 20 iterations. These settings may be adapted according to the specific editing scenario to optimal performance.

## A.3 User Study Details.

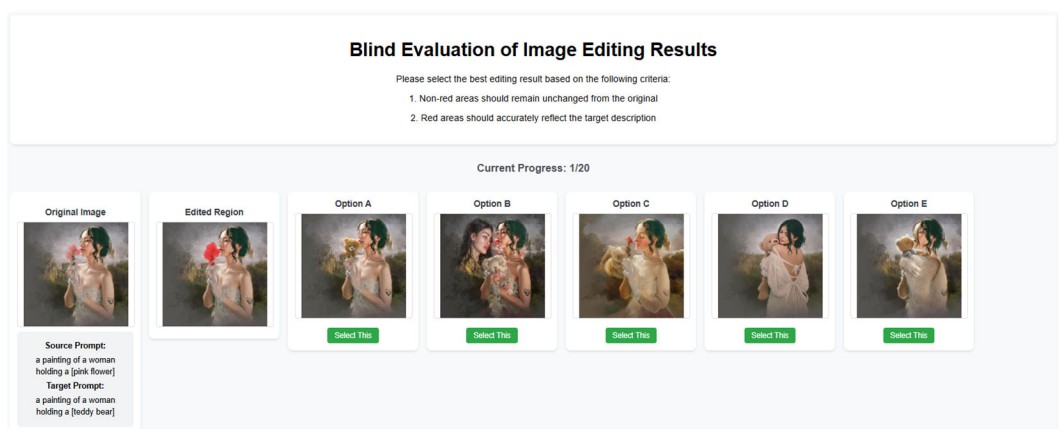

Figure 10: Custom web interface of user study.

Each volunteers is asked to select the best editing result via a custom web interface specifically developed for this evaluation, as shown in Figure 10. The interface presents a source image along with its corresponding prompt, the edited region, a target prompt, and five edited results. The methods behind these results are anonymized and displayed in a randomized order for each evaluation.

## A.4 More Ablation Study.

**Ablation on Transformer LoRA.** As shown in Table 8, applying LoRA solely to FFN layers yields a more favorable trade-off between background preservation and text alignment compared to other configurations. Therefore, we adopt this configuration in our final design, enabling effective editing with minimal additional parameter overhead.

Table 8: Ablation on Transformer LoRA. Attn denotes both self-attention and cross-attention.

| FFN | Attn | Background Preservation | | | | Text Alignment | | |
|-----|------|-------|---------|---------|---------|--------|--------|--------|
| | | PSNR↑ | $\text{LPIPS}_{10^3}$↓ | $\text{MSE}_{10^4}$↓ | $\text{SSIM}_{10^2}$↑ | Whole↑ | Edited↑ | $\text{IR}_{\times 10}$↑ |
| ✓ | ✗ | 28.50 | 31.58 | 22.94 | 92.36 | **26.22** | **23.99** | **5.39** |
| ✗ | ✓ | 28.50 | 31.11 | 23.07 | 92.32 | 25.72 | 23.34 | 3.93 |
| ✓ | ✓ | **28.81** | **30.31** | **21.39** | **92.64** | 25.61 | 23.29 | 4.23 |

**Ablation on Piecewise Linear Smoothing Kernel.** Table 2 in the main body only presents better balance results for Gaussian and linear kernels. The full results under varying hyperparameter configurations of Gaussian and linear kernels are provided in Tables 9 and 10, respectively. In the case of Gaussian kernel, increasing $\alpha$ enlarges the smooth transition zone, compromising background retention while improving text alignment. In the case of the linear kernel, when fixing $\tau_1$ and gradually increasing $\tau_2$, the transition zone width of the linear kernel ($\tau_2 - \tau_1$) increases accordingly. This leads to improved text alignment metrics but at the cost of degraded background preservation performance. Conversely, when $\tau_2$ is fixed and $\tau_1$ increases, both text alignment and background preservation tend to deteriorate. These observations indicate that $\tau_1$ and $\tau_2$ play a critical role in balancing edit fidelity

and content preservation. Overall, the linear kernel setting of $\tau_1 = 1$ and $\tau_2 = 4$ offers a better trade-off, achieving strong text alignment (e.g., $IR_{10} = 5.39$) while keeping background distortion (e.g., LPIPS = 31.58) within acceptable limits.

Table 9: Quantitative results of ablating the Gaussian kernel.

| $\alpha$ | Background Preservation | | | | Text Alignment | | |
|---|---|---|---|---|---|---|---|
| | PSNR↑ | LPIPS$_{10^3}$↓ | MSE$_{10^4}$↓ | SSIM$_{10^2}$↑ | Whole↑ | Edited↑ | IR$_{10}$↑ |
| 1 | **29.40** | **28.59** | **18.45** | **93.01** | 26.09 | 23.79 | 4.74 |
| 2 | 28.63 | 31.16 | 21.71 | 92.45 | 26.08 | 23.71 | 4.66 |
| 3 | 28.15 | 32.91 | 24.40 | 92.17 | 26.10 | **23.81** | 4.61 |
| 4 | 28.08 | 33.05 | 25.00 | 92.25 | **26.23** | 23.73 | **4.91** |

Table 10: Quantitative results of ablating the linear kernel.

| $\tau_1$ | $\tau_2$ | Background Preservation | | | | Text Alignment | | |
|---|---|---|---|---|---|---|---|---|
| | | PSNR↑ | LPIPS$_{10^3}$↓ | MSE$_{10^4}$↓ | SSIM$_{10^2}$↑ | Whole↑ | Edited↑ | IR$_{10}$↑ |
| 0 | 1 | **29.19** | **29.15** | **19.47** | **92.87** | 26.13 | 23.74 | 4.83 |
| 0 | 2 | 29.16 | 29.23 | 19.48 | 92.86 | 26.11 | 23.86 | 5.04 |
| 0 | 3 | 28.92 | 29.87 | 20.35 | 92.79 | **26.24** | 23.85 | 5.00 |
| 0 | 4 | 28.69 | 30.60 | 21.38 | 92.66 | 26.19 | **24.01** | 4.79 |
| 0 | 5 | 28.45 | 31.68 | 22.86 | 92.51 | 26.20 | 23.80 | 4.71 |
| 1 | 2 | 28.80 | 30.21 | 22.34 | 92.65 | 26.07 | 23.71 | 4.72 |
| 1 | 3 | 28.46 | 31.75 | 22.54 | 92.35 | 26.13 | 23.83 | 4.85 |
| 1 | 4 | 28.50 | 31.58 | 22.94 | 92.36 | 26.22 | 23.99 | **5.39** |
| 1 | 5 | 28.41 | 31.74 | 22.98 | 92.36 | 26.14 | 23.83 | 4.73 |
| 2 | 3 | 28.50 | 31.44 | 22.46 | 92.50 | 26.18 | 23.53 | 4.96 |
| 2 | 4 | 28.14 | 32.69 | 24.47 | 92.31 | 26.17 | 23.72 | 4.91 |
| 2 | 5 | 27.53 | 36.17 | 29.65 | 91.79 | 26.21 | 23.74 | 5.12 |

Table 11: **Quantitative results of ablating the mask.** EditInfinity-u denotes user-provided masks; EditInfinity-c denotes cross-attention masks. Best and second-best results are shown in **bold** and *italics*.

| Method | Base Model | Background Preservation | | | | Text Alignment | | |
|---|---|---|---|---|---|---|---|---|
| | | PSNR↑ | LPIPS$_{10^3}$↓ | MSE$_{10^4}$↓ | SSIM$_{10^2}$↑ | Whole↑ | Edited↑ | IR$_{10}$↑ |
| P2P[16] | | 18.81 | 197.11 | 197.69 | 73.68 | 25.10 | 22.98 | 0.29 |
| MasaCtrl[4] | | 23.36 | 95.45 | 77.63 | 81.88 | 23.30 | 20.92 | -3.82 |
| P2P-Zero[31] | U | 20.92 | 161.28 | 137.64 | 77.02 | 22.89 | 21.09 | -5.71 |
| NTI[29] | | *28.08* | 57.94 | 36.10 | 85.17 | 24.71 | 22.51 | 3.63 |
| PnP-Inv[20] | | 23.60 | 103.12 | 72.77 | 81.11 | 25.05 | 22.94 | 3.34 |
| NP[28] | | 27.24 | 62.40 | 37.79 | 84.92 | 24.89 | 22.67 | 2.92 |
| StableFlow[2] | T | 23.68 | 72.77 | 78.61 | 88.11 | 23.17 | 21.21 | 0.76 |
| RF-Edit[53] | | 27.26 | 92.27 | *34.46* | 86.67 | 24.65 | 22.03 | 0.61 |
| *EditInfinity-c* | A | 27.47 | *44.97* | 46.91 | *90.30* | *25.71* | *23.22* | **5.40** |
| **EditInfinity-u** | | **28.50** | **31.58** | **22.94** | **92.36** | **26.22** | **23.99** | *5.39* |

**Ablation on Mask.** Our method assumes the user provides masks. Indeed, this is a well-established task setting in image editing [37, 59], especially when text alone is insufficient for the precise localization of the user-desired editing region. This challenge of accurately conveying user intent has long been recognized in controllable image generation. To enhance controllability, ControlNet [59] leverages visual priors such as edge maps, while DreamBooth [37] utilizes user-provided images to capture detailed features not easily conveyed by text.

While our method assumes user-provided masks by default, it can also leverage Infinity's cross-attention maps [16] for automatic mask generation without modifying the framework. Table 11 reports comprehensive comparisons and shows that our method is not highly sensitive to the source of the mask—strong performance is achieved in both cases.

**Ablation on Multi-scale Autoregressive Editing.** By blending source tokens at each scale in an autoregressive (AR) manner, our method provides continuous guidance for editing region generation at subsequent scales. In contrast, the non-autoregressive (NAR) approach blends source tokens only at the end of each scale, without influencing the token generation process at the next scale. This leads to incoherent transitions and visually inconsistent edits, as illustrated in Figure 11. Table 12 further supports this observation: AR consistently outperforms NAR in both background preservation and text alignment. These results highlight the necessity of autoregressive guidance for achieving harmonious and realistic edits.

Table 12: Quantitative results of multi-scale autoregressive editing.

| Blend | Background Preservation | | | | Text Alignment | | |
|---|---|---|---|---|---|---|---|
| | PSNR↑ | $LPIPS_{10^3}$↓ | $MSE_{10^4}$↓ | $SSIM_{10^2}$↑ | Whole↑ | Edited↑ | $IR_{10}$↑ |
| NAR | 25.50 | 42.59 | 38.39 | 91.00 | 25.98 | 23.64 | 3.54 |
| AR | **28.50** | **31.58** | **22.94** | **92.36** | **26.22** | **23.99** | **5.39** |

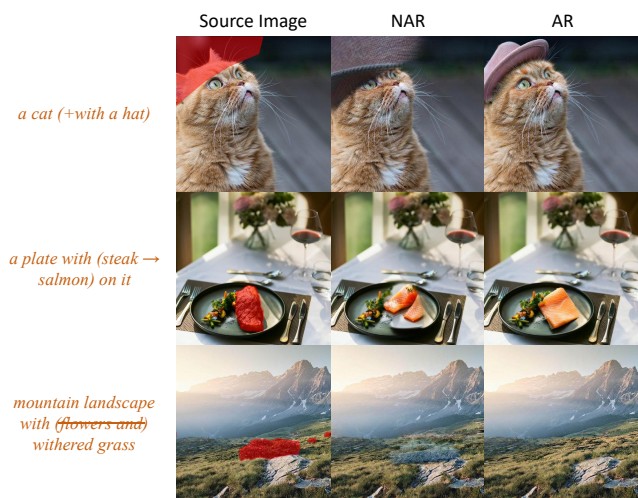

Figure 11: Illustrations of ablating the Multi-scale Autoregressive Editing.

## A.5 Additional Qualitative Results.

We present additional qualitative results to further demonstrate the effectiveness of our method, as shown in Figure 12 and 13. These results include diverse editing types across add object, change object, delete object, change content, change pose, change color, change background, change material, and change style. We also provide comparisons with state-of-the-art methods, highlighting our model's ability to preserve background details and align with target prompts in image editing.

## A.6 Broader Impacts

Our proposed method enables high-quality image editing. Positive societal impacts include its potential applications in education (e.g., visual content adaptation for learning) and creative industries (e.g., graphic design and media production). However, potential negative societal impacts include misuse for deceptive content creation (e.g., deepfakes or misinformation). We acknowledge the dual-use nature of image generation technologies and emphasize responsible deployment.

### A.7 Safeguards

To mitigate risks associated with misuse, we adopt the following safeguards:

- We will release the model under a research-use-only license.
- Model checkpoints and code will include a usage agreement that prohibits harmful or deceptive use cases (e.g., unauthorized alteration of real people's images).
- All datasets used for editing are publicly available and contain no private or personally identifiable information.

### A.8 Ethical Considerations

There is no potential risks incurred by study participants in this paper. As such, Institutional review board (IRB) approval was not required.

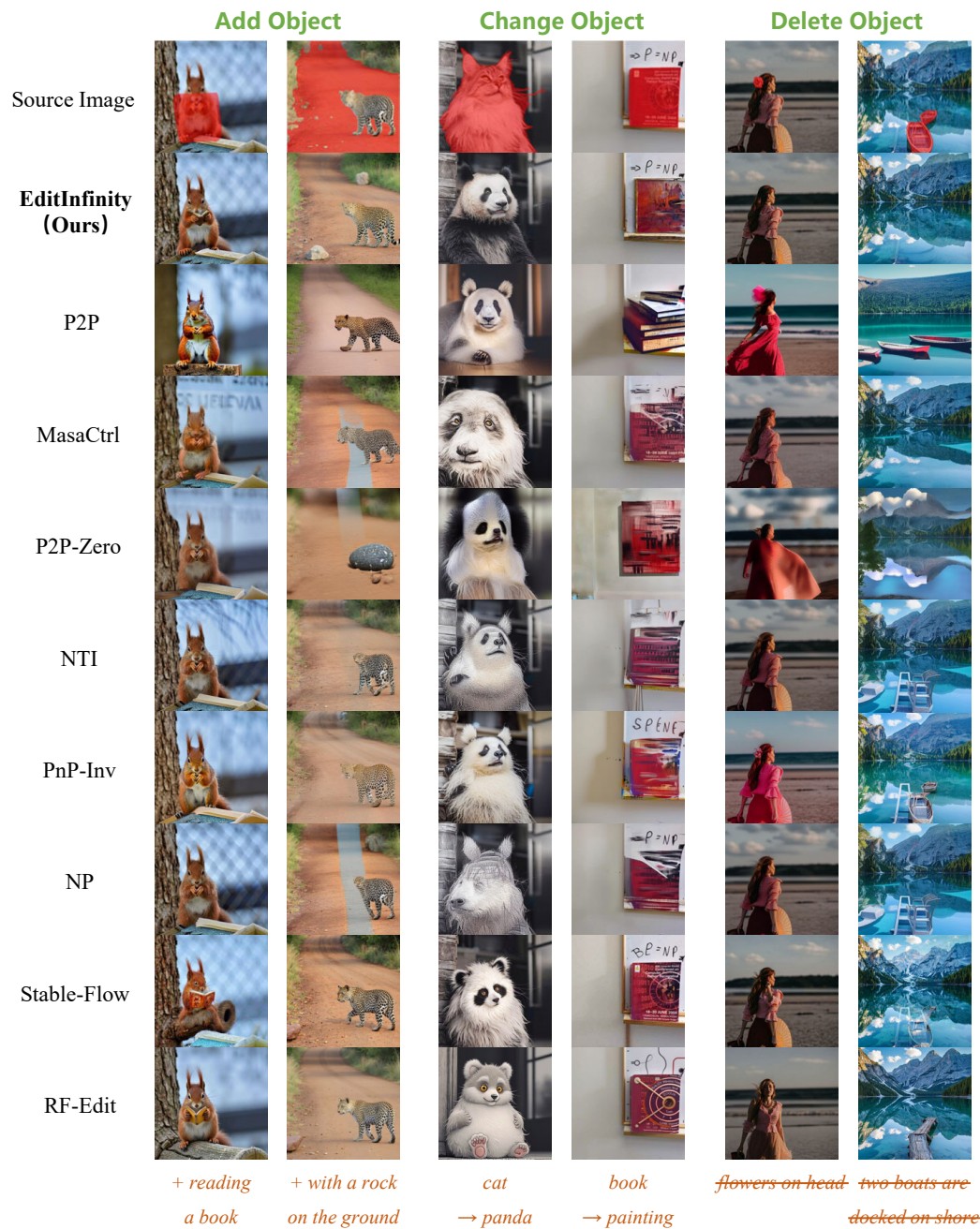

Figure 12: **Qualitative results on PIE-Bench across add, change, and delete object.** The red mask denotes the edited region $M$, expected to follow the prompt, while other regions retain the background.

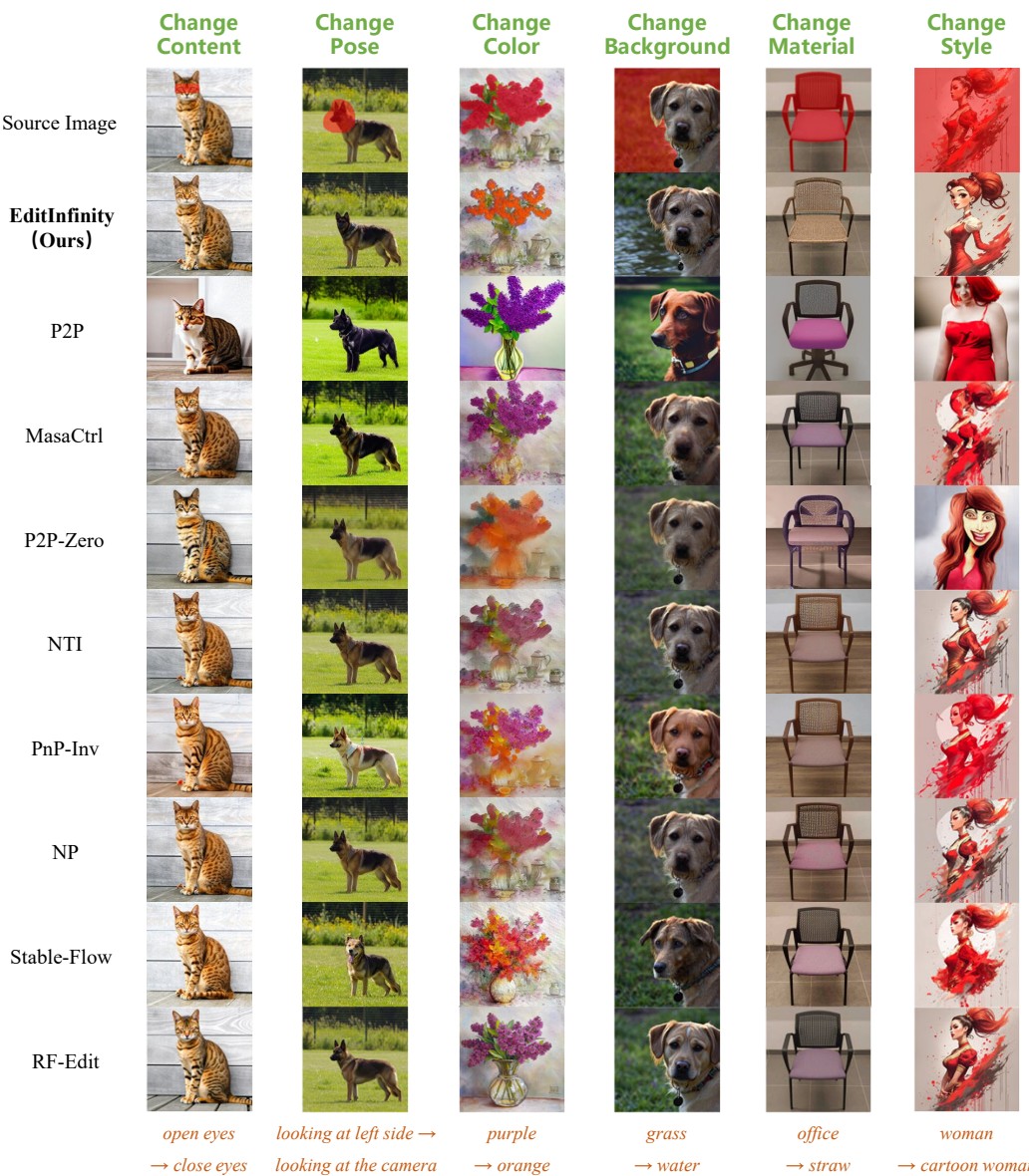

Figure 13: **Qualitative results on PIE-Bench across change content, change pose, change color, change background, change material, and change style.** The red mask denotes the edited region $M$, expected to follow the prompt, while other regions retain the background.

