# OpenReview forum: "EditInfinity: Image Editing with Binary-Quantized Generative Models"
_NeurIPS.cc/2025/Conference — NeurIPS 2025 poster_

### Official Review · Reviewer_SEN3 · 2025-06-25

**Clarity:** 2
**Significance:** 3
**Originality:** 3
**Rating:** 5
**Confidence:** 4

**Summary:**

This paper introduces EditInfinity - an editing model built on the top of Infinity T2I. It offers the advantages compared to diffusion model because VQ based generative model has the ground truth of the intermediate outputs. It learns the learnable prompt and LORA for text prompt rectification and image style preservation, and propose a smoothing method through piecewise linear kernel and multi-scale autoregressive. It freezes the original Infinity to have minimal training costs.

**Questions:**

1/ Please add more comparisons with top models like gpt-4o or Gemini.
2/ Better ablation study for multi-scale autoregressive editing instead of putting one visualization example
3/ Better write-up for method section 4, including a/ better definition of symbols, b/ training strategy (stage 1, stage 2, ....)

**Ethical Concerns:**

["NO or VERY MINOR ethics concerns only"]

**Final Justification:**

After reading all rebuttals and other review feedback, I am leaning to weak accept to accept. VQ based generation models are quite behind diffusion one, and this work achieves solid results across multiple edit tasks compared to diffusion base models. I think this work can be a good milestone towards VQ based editing work.

**Limitations:**

yes

**Paper Formatting Concerns:**

No formatting issues.

**Quality:**

3

**Strengths And Weaknesses:**

Strengths
* This model is one of first few AR model which can have editing capabilities with great quality - this is the top strength to me
* It freezes infinity model so the method is very efficient.
* Some improvements for alignment and style preservation through learnable prompts and LORA
* Blended style multi-style AR is interesting.
* Results are good based on benchmark scores and user study

Weakness
* It missed some top editing models in close source like Gpt-4o and gemini for comparisons.
* Ablation quality could be improved for multi-scale autoregressive editing. It only has one visualization example and it is difficult to know whether it is cherry picked.
* <minor> It is better have a definition of all symbols in Figure 2 and Figure 3, as well as training strategy (stage 1, stage 2, ....).

---

> ### Author Rebuttal · Authors · 2025-07-30
>
> Thank you for your insightful comments. We respond to your concerns point by point as follows.
>
> **Main Comment**
>
> > **Q1: Please add more comparisons with top models like gpt-4o or Gemini.**
>
> **A1:** We sincerely thank the reviewer for pointing out this valuable concern. As shown in Table 1 below, we provide a comparison with the **closed-source Gemini**, which serves as a strong commercial baseline. Due to time constraints, we are not able to include results for GPT-4o. Nonetheless, we believe the comparison with Gemini provides a representative and meaningful reference point for assessing our method. As shown in Table 1, the thoughtful design of our method enables it to outperform even the closed-source Gemini.
>
> **Table 1: Quantitative results of multi-scale autoregressive editing.**
> | Method              | PSNR↑ | LPIPS(×10³)↓ | MSE(×10⁴)↓ | SSIM(×10²)↑ | Whole↑ | Edited↑ | IR(×10)↑ |
> |---------------------|-----------|------------------|----------------|-----------------|-----------|------------|-----------|
> | P2P                  | 17.87     | 208.80           | 219.88         | 71.14           | 25.01     | 22.44      | 0.017     |
> | MasaCtrl          | 22.17     | 106.62           | 86.97          | 79.67           | 23.96     | 21.16      | -1.66     |
> | P2P-Zero         | 20.44     | 172.22           | 144.12         | 74.67           | 22.80     | 20.54      | -6.59     |
> | NTI                   | 27.03     | 60.67            | 35.86          | 84.11           | 24.75     | 21.86      | 2.77      |
> | PnP-Inv            | 22.46     | 106.06           | 80.45          | 79.68           | 25.41     | 22.62      | 4.17      |
> | NP                    | 26.21     | 69.01            | 39.73          | 83.40           | 24.61     | 21.87      | 2.42      |
> | StableFlow       | 21.64     | 92.28            | 115.21         | 84.94           | 24.65     | 21.70      | 1.88      |
> | RF-Edit             | 23.22     | 131.18           | 75.00          | 81.44           | 25.22     | 22.40      | 5.18      |
> | Gemini              |   23.22   |  105.17       |   188.63         |    81.10        |   25.28     |   22.28     |   5.30     |
> | **EditInfinity (Ours)**   | **27.95** | **33.08**        | **24.27**      | **92.12**       | **26.41** | **23.47**  | **5.88**  |
>
>
> > **Q2: Better ablation study for multi-scale autoregressive editing instead of putting one visualization example.**
>
> **A2:** We thank the reviewer for this insightful question. Due to space limitations in the main paper, this has been discussed in Table 4 of the supplementary material.
>
> **Table 4: Quantitative results of multi-scale autoregressive editing.**
>
> | Blend       | PSNR↑ | LPIPS(×10³)↓ | MSE(×10⁴)↓ | SSIM(×10²)↑ | Whole↑ | Edited↑ | IR(×10)↑ |
> |-------|--------|----------------|--------------|----------------|---------|----------|---------|
> | NAR   | 25.50  | 42.59          | 38.39        | 91.00          | 25.98   | 23.64    | 3.54    |
> | **AR**| **28.50**  | **31.58**        | **22.94**      | **92.36**        | **26.22** | **23.99**  | **5.39**  |
>
> By blending source tokens at each scale in an autoregressive (AR) manner, our method provides continuous guidance for editing region generation at subsequent scales.  In contrast, the non-autoregressive (NAR) approach blends source tokens only at the end of each scale, without influencing the token generation process at the next scale. This leads to incoherent transitions and visually inconsistent edits, as illustrated in Figure 9 of main paper and Figure 2 of supplementary material. Table 4 here further supports this observation: AR consistently outperforms NAR in both background preservation and text alignment. These results highlight the necessity of autoregressive guidance for achieving harmonious and realistic edits.
>
> > **Q3: Better write-up for method section 4, including a/ better definition of symbols, b/ training strategy (stage 1, stage 2, ....)**
>
> **A3:** We sincerely thank the reviewer for pointing out the need for clearer symbol definitions in Figures 2 and 3, as well as a more structured presentation of the training strategy in Section 4.
>
> (a) We have added dedicated legend areas to Figures 2 and 3 to explain the meaning of all relevant symbols, such as $t_{sou}, t_{ins}, t_{l}, R_{1...K}^{sou}, R_{1...K}^{inv}, R_{1...K}^{tar}, E_{1...K}^{tar}, \mathcal{L}_{inv}, \mathcal{G}$, etc. As the rebuttal does not support figures, we are unable to show the updated figures here.
>
> (b) We have clearly annotated Stage 1 and Stage 2 in Figure 2 and provided a more precise and structured definition of both stages in Section 4, as outlined below.
>
> - Image inversion stage 1: textual prompting rectification.
> - Image inversion stage 2: image style preservation.
>
>  We greatly appreciate your suggestions, which will help improve the clarity and rigor of our method section.

---

> > ### Comment · Reviewer_SEN3 · 2025-08-05
> >
> > Thank you for the thoughtful rebuttal and the authors’ detailed responses. The comparison with Gemini strengthens the validity of the results. This is an excellent contribution to an underexplored area in VQ-based editing, and I believe it will attract interest from researchers in the field. I will maintain my positive score.

---

### Official Review · Reviewer_Ghm8 · 2025-06-29

**Clarity:** 4
**Significance:** 3
**Originality:** 2
**Rating:** 4
**Confidence:** 3

**Summary:**

**EditInfinity** introduces a novel text-driven image editing method that leverages binary-quantized generative models as an alternative to traditional diffusion-based approaches. By utilizing discrete token representations, the method enables precise image inversion with exact supervision and avoids the approximation errors typical of diffusion inversion. The authors enhance this process through a learnable prompt mechanism and LoRA-based style preservation. For the editing phase, they propose a holistic smoothing strategy that blends source and target tokens using a piecewise linear kernel, allowing for seamless and localized modifications. Evaluated on the PIE-Bench benchmark, EditInfinity achieves state-of-the-art results in both background preservation and semantic alignment, significantly outperforming diffusion-based baselines across a variety of editing tasks.

**Questions:**

1. How well does the method handle vague or high-level prompts that lack explicit object references or structure?
2. The inversion and editing stages seem computationally demanding. What is the latency for a single edit?
3. How is the edited region determined in practical scenarios? Can the method infer edit regions automatically based on differences between source and target prompts, or does it rely entirely on manual mask annotation?

**Ethical Concerns:**

["NO or VERY MINOR ethics concerns only"]

**Final Justification:**

Thank authors for the clarification and effort, and I have decided to increase my rating. But the method can be further improved with more flexible setting and better efficiency.

**Limitations:**

Please refer to the weaknesses section.

**Quality:**

3

**Strengths And Weaknesses:**

### Strengths

1. **Clear Motivation and Novelty**: The paper addresses limitations of diffusion-based editing, especially inversion error, and innovatively adapts a VQ-based model, which is a less explored direction.
2. **Thorough Evaluation**: The authors conduct extensive quantitative and qualitative experiments, including a user study, to demonstrate improvements across various editing tasks. EditInfinity outperforms existing methods in key metrics, especially in preserving unedited regions while aligning edits with prompts.

### Weaknesses

1. **Structured Editing Setup**: The method heavily depends on well-aligned source-target prompt pairs and user-provided region masks. This reliance on structured inputs may limit its generalizability to open-ended or ambiguous editing tasks where such precise supervision is unavailable.
2. **Computation Cost**: The framework involves several non-trivial steps, including prompt optimization, LoRA fine-tuning, and multi-scale autoregressive token generation. This complexity could hinder real-time applications and make the method less practical.
3. **Manual Design**: The framework includes several manually crafted components, such as user-defined editing masks and a fixed smoothing kernel. These elements introduce human priors into the pipeline, and it remains unclear how robust the approach is to noisy inputs or complex, less structured scenes.

---

> ### Author Rebuttal · Authors · 2025-07-30
>
> We sincerely appreciate your thoughtful and constructive feedback. Below, we respond to each of your comments in detail and have incorporated the corresponding revisions into the final version of the paper.
>
> **Main Comment**
>
> > **Q1: How well does the method handle vague or high-level prompts that lack explicit object references or structure?**
>
> **A1:** We sincerely thank you for highlighting this concern. We provide a detailed response from the following two perspectives: ’Prompt Understanding‘ and ’Open-ended Editing Tasks‘.
>
> **(a) Prompt Understanding.** Our method is designed to operate with known editing regions and does not focus on prompt understanding. In scenarios involving vague or high-level prompts without clear object references, external multimodal large language models (MLLMs) may be integrated to enhance prompt understanding.
>
> **(b) Open-ended Editing Tasks.** Given that PIE-Bench contains nearly 50% natural images, it already serves as a comprehensive benchmark for assessing our method. However, to specifically evaluate complex scenes, we select 20 MagicBrush images (due to time constraints) filtered by GPT-4o, comprising five samples each with 2, 3, 4, and 5 primary objects. Table 1 demonstrates the superiority of our method in handling complex scenes.
>
> **Table 1: Quantitative results on the complex scene images from MagicBrush .**
> |Method|PSNR↑|LPIPS(×10³)↓|MSE(×10⁴)↓|SSIM(×10²)↑|Whole↑|Edited↑|
> |-|-|-|-|-|-|-|
> |NTI|8.81|452.03|1380.48|39.63|19.90|16.83|
> |RF-Edit|26.00|121.84|33.98|84.73|24.13|18.29|
> |**Ours**|**31.23**|**24.30**|**9.90**|**91.70**|**24.19**|**20.07**|
>
> Note: Due to time constraints, we select representative methods with strong performance from both diffusion U-Net-based models and diffusion transformer-based models for comparison.
>
> > **Q2: The inversion and editing stages seem computationally demanding. What is the latency for a single edit?**
>
> **A2:** We conduct a runtime comparison of our method and other methods on a single NVIDIA L20 GPU, measuring both inversion and editing time, as shown in Table 2 below. A key advantage lies in its efficiency during multiple edits on one image—a common real-world scenario. Once the inversion for a given image is completed, subsequent edits can be performed within **3.64 seconds**—over **7× faster** than other methods on average (initial inversion time only 4× longer than other methods on average). This design effectively front-loads the computational cost.
>
>
> **Table 2: Runtime comparison of inversion and editing.**
> |Method|Inversion|Once Editing|Twice Editing|
> |-|-|-|-|
> |P2P|14.40s|10.28s|10.28s|
> |MasaCtrl|**5.19s**|17.45s|17.45s|
> |P2P-Zero|13.31s|62.29s|62.29s|
> |NTI|95.54s|10.32s|10.32s|
> |PnP-Inv|8.32s|9.54s|9.54s|
> |NP|9.00s|10.37s|10.37s|
> |StableFlow|13.85s|27.20s|27.20s|
> |RF-Edit|55.48s|54.07s|54.07s|
> |**Ours**|107.06s|**3.64s**|**3.64s**|
>
> > **Q3: How is the edited region determined in practical scenarios? Can the method infer edit regions automatically based on differences between source and target prompts, or does it rely entirely on manual mask annotation?**
>
> **A3:** We appreciate this insightful question. Our method assumes the user provides masks. Indeed, this is a well‑established task setting in image editing [1] [2], especially when text alone is insufficient for the precise localization of the user-desired editing region. This challenge of accurately conveying user intent has long been recognized in controllable image generation. To enhance controllability, ControlNet [3] leverages visual priors such as edge maps, while DreamBooth [4] utilizes user-provided images to capture detailed features not easily conveyed by text.
>
> While our method assumes user-provided masks by default, it can also leverage Infinity’s cross-attention maps [5] for **automatic mask generation** without modifying the framework. Specifically, we automatically align differing words $x$ between the source and target prompts. After completing inversion, we input the source or target prompt containing $x$ into Infinity and extract the cross-attention map corresponding to $x$. A threshold is then applied: values above the threshold are set to 0 (mask foreground), and others to 1 (background). Table 3 shows that our method is not highly sensitive to the source of the mask—strong performance is achieved in both cases. We will include further discussion on this aspect in the final version.
>
> **Table 3: Quantitative results on the random class of PIE-Bench.** Ours-u denotes user-provided masks; Ours-c denotes cross-attention masks. Best and second-best results are shown in **bold** and *italics*.
> |Method|PSNR↑|LPIPS(×10³)↓|MSE(×10⁴)↓|SSIM(×10²)↑|Whole↑|Edited↑|IR(×10)↑|
> |-|-|-|-|-|-|-|-|
> |P2P|18.81|197.11|197.69|73.68 |25.10|22.98|0.29|
> |MasaCtrl|23.36|95.45|77.63|81.88|23.30|20.92|-3.82|
> |P2P-Zero|20.92|161.28|137.64|77.02|22.89|21.09|-5.71|
> |NTI|*28.08*|57.94|36.10|85.17|24.71|22.51|3.63|
> |PnP-Inv|23.60|103.12|72.77|81.11|25.05|22.94|3.34|
> |NP|27.24|62.40|37.79| 84.92|24.89|22.67|2.92|
> |StableFlow|23.68|72.77|78.61|88.11|23.17|21.21|0.76|
> |RF-Edit|27.26| 92.27| *34.46*|86.67| 24.65| 22.03|0.61|
> |*Ours-c*|27.47|*44.97*|46.91|*90.30*|*25.71*|*23.22*|**5.40**|
> |**Ours-u**|**28.50**|**31.58**|**22.94**|**92.36**|**26.22**|**23.99**|*5.39*|
>
> Note: Due to time constraints, experiments are conducted on the “random class” of PIE-Bench, which includes all editing types and supports efficient and unbiased evaluation of model components.
>
> [1] Nitzan Y, et al. Lazy diffusion transformer for interactive image editing[C]//ECCV. 2024.
>
> [2] Zhuang J, et al. A task is worth one word: Learning with task prompts for high-quality versatile image inpainting[C]//ECCV. 2024.
>
> [3] Zhang L, et al. Adding conditional control to text-to-image diffusion models[C]//ICCV. 2023.
>
> [4] Ruiz N, Li Y, et al. Dreambooth: Fine tuning text-to-image diffusion models for subject-driven generation[C]//CVPR. 2023.
>
> [5] Amir Hertz, et al. Prompt-to-Prompt Image Editing with Cross-Attention Control[C]//ICLR. 2023.

---

### Official Review · Reviewer_LWWe · 2025-06-30

**Clarity:** 3
**Significance:** 4
**Originality:** 3
**Rating:** 4
**Confidence:** 4

**Summary:**

This paper introduces EditInfinity, which is a text-based image editing method based on the autoregressive text-to-image generation model Infinity, which succeeds in high quality generation among vector quantized models available. The authors motivate the paper from the existing inversion-then-editing approach, where instead of inverting into a noisy latent (like in diffusion models), they initially invert the input text prompt by an extended embedding representation, which is followed by LoRA based adaptation for the reconstruction of precise details. Given a target text prompt, an input image and a region of interest specified by a mask, EditInfinity performs editing by manipulating the multi-scale quantized feature maps with a smoothed mask $\mathcal{G}$, then reconstruct the image using these manipulated feature maps. Authors demonstrate the effectiveness of the approach by using PIE-Bench, where they compare their approach with diffusion/flow-matching based editing methods. The provided quantitative analyses and qualitative results shows the effectiveness of the method.

**Questions:**

- How are the masks used in the editing process obtained in the evaluations? Does the method assume that these masks are provided by the user or is there an automated pipeline to achieve this?
- Is the method able to achieve disentanglement for un-maskable edits such as facial attribute change?
- What are the failure cases of the method in detail?
- Is there are timing constraint on the method, as a two stage optimization is required for editing?

**Ethical Concerns:**

["NO or VERY MINOR ethics concerns only"]

**Final Justification:**

The rebuttal addressed my questions regarding the mask sensitivity, identity preservation, performance on "on-the-wild" images and evaluation details. After reviewing the author rebuttal and the other reviews, I am leaning towards positive.

**Limitations:**

- The limitations are very briefly addresses over a small discussion. I believe that further details should be included like failure cases as discussed over the *Questions* and *Strengths And Weaknesses* sections. In addition the authors are encouraged to include a timing based comparison due to the two stage optimization that the method requires (for each editing example). For further comments on suggested improvements over the limitations, please refer to *Questions* and *Strengths And Weaknesses* sections.

**Paper Formatting Concerns:**

The paper is adequately formatted.

**Quality:**

3

**Strengths And Weaknesses:**

**Strengths**
- The paper tackles the image editing problem using autoregressive image editing methods with a new inversion scheme which mainly depends on approximating the ideal generation prompt, and minor adjustments on the autoregressive model with LoRA-based optimization.
- The proposed method achieves succeeds in background preservation compared to existing approaches
- The editing pipeline proposed provides satisfactory results on edits that can be constrained with a region identified by a mask.
- Authors provide ablations for the design decisions made for each of the components of the method.

**Weaknesses**
- The limitations of the method is discussed very briefly, where the authors should provide more qualitative examples for the failure cases.
- The edits that can be performed with the proposed approach seems limited with an input mask, where providing the whole image as the editing region of interest seems to be disturbing the background preservation properties and the preservation of the input pose, structure (see Fig. 6, row 9). If there is such limitation, the authors should explicitly discuss this for transparency.
- There are missing details in the quantitative evaluation setup and method details. For the details on the method, the proposed approach involves two-stage training per image. To fairly assess if this approach is feasible as an editing method or not, the authors should provide a time comparison with the competing approach in terms of the overhead on top of the base model. In addition, the evaluation requires an input mask for EditInfinity, where the method used to obtain these masks are not described in detail. The authors should privide such details on the evaluation setup.
- Given the short description about the limitations from the authors, the proposed method succeeds in masked edits. While this is a valid use case for editing, it does not cover a majority of the edits to be performed. As an example, editing facial attributes is nor always maskable (e.g. adding smile). It is not clear if the method can perform such edits or not, if it can there should be examples on such detailed cases. In addition, the editing examples provided are rather simplistic cases where there is a single subject that is centered in the image. This makes the performance in on-the-wild images questionable. The authors should provide examples for such use cases.

---

> ### Author Rebuttal · Authors · 2025-07-30
>
> Thanks for your careful and valuable comments. We will explain your concerns point by point.
>
> **Main Comment**
>
> > **Q1: How are the masks used in the editing process obtained in the evaluations? Does the method assume that these masks are provided by the user or is there an automated pipeline to achieve this?**
>
> **A1:** We appreciate this insightful question. Our method assumes the user provides masks. Indeed, this is a well‑established task setting in image editing [1] [2], especially when text alone is insufficient for the precise localization of the user-desired editing region. This challenge of accurately conveying user intent has long been recognized in controllable image generation. To enhance controllability, ControlNet [3] leverages visual priors such as edge maps, while DreamBooth [4] utilizes user-provided images to capture detailed features not easily conveyed by text.
>
> While our method assumes user-provided masks by default, it can also leverage Infinity’s cross-attention maps [5] for automatic mask generation without modifying the framework. Specifically, we automatically align differing words $x$ between the source and target prompts. After completing inversion, we input the source or target prompt containing $x$ into Infinity and extract the cross-attention map corresponding to $x$. A threshold is then applied: values above the threshold are set to 0 (mask foreground), and others to 1 (background). Table 1 shows that our method is not highly sensitive to the source of the mask—strong performance is achieved in both cases. We will include further discussion on this aspect in the final version.
>
> **Table 1: Quantitative results on the random class of PIE-Bench.** Ours-u denotes user-provided masks; Ours-c denotes cross-attention masks. Best and second-best results are shown in **bold** and *italics*.
> |Method|PSNR↑|LPIPS(×10³)↓|MSE(×10⁴)↓|SSIM(×10²)↑|Whole↑|Edited↑|IR(×10)↑|
> |-|-|-|-|-|-|-|-|
> |P2P|18.81|197.11|197.69|73.68 |25.10|22.98|0.29|
> |MasaCtrl|23.36|95.45|77.63|81.88|23.30|20.92|-3.82|
> |P2P-Zero|20.92|161.28|137.64|77.02|22.89|21.09|-5.71|
> |NTI|*28.08*|57.94|36.10|85.17|24.71|22.51|3.63|
> |PnP-Inv|23.60|103.12|72.77|81.11|25.05|22.94|3.34|
> |NP|27.24|62.40|37.79| 84.92|24.89|22.67|2.92|
> |StableFlow|23.68|72.77|78.61|88.11|23.17|21.21|0.76|
> |RF-Edit|27.26| 92.27| *34.46*|86.67| 24.65| 22.03|0.61|
> |*Ours-c*|27.47|*44.97*|46.91|*90.30*|*25.71*|*23.22*|**5.40**|
> |**Ours-u**|**28.50**|**31.58**|**22.94**|**92.36**|**26.22**|**23.99**|*5.39*|
>
> Note: Due to time constraints, experiments are conducted on the “random class” of PIE-Bench, which includes all editing types and supports efficient and unbiased evaluation of model components.
>
> [1] Nitzan Y, et al. Lazy diffusion transformer for interactive image editing[C]//ECCV. 2024.
>
> [2] Zhuang J, et al. A task is worth one word: Learning with task prompts for high-quality versatile image inpainting[C]//ECCV. 2024.
>
> [3] Zhang L, et al. Adding conditional control to text-to-image diffusion models[C]//ICCV. 2023.
>
> [4] Ruiz N, Li Y, et al. Dreambooth: Fine tuning text-to-image diffusion models for subject-driven generation[C]//CVPR. 2023.
>
> [5] Amir Hertz, et al. Prompt-to-Prompt Image Editing with Cross-Attention Control[C]//ICLR. 2023.
>
> > **Q2: Is the method able to achieve disentanglement for un-maskable edits such as facial attribute change?**
>
> **A2:** We thank the reviewer for this insightful question.
>
> **(a) Facial attribute change.** Yes, our method can realize facial attribute change. **PIE-Bench**, used for evaluation in main paper, is a comprehensive image editing benchmark covering natural human-centric scenarios. We use GPT-4o to filter editing instructions, resulting in 16 representative examples such as **'change the expression from laughing to angry'** and 'add the age of the woman'. Our method performs favorably on these cases. Please refer to Table 1 in the main paper for qualitative results, given image upload constraints.
> ```
> |-- random
> |-- change_object
>      |-- natural*
>           |-- animal
>           |-- human*
>           |-- indoor
>           |-- outdoor
>      |-- artificial
>           |-- ...
> |-- add_object
>      |-- ...
> |-- delete_object
>      |-- ...
> |-- change_attribute_content
>      |-- ...
> |-- change_attribute_pose
>      |-- ...
> |-- change_attribute_color
>      |-- ...
> |-- change_attribute_material
>      |-- ...
> |-- change_background
>      |-- ...
> |-- change_style
>      |-- ...
> ```
> Furthermore, we randomly select 20 face images from **FFHQ** (due to time constraints) to perform unmasked facial attribute changes, including **age, expression, skin tone**, and more. Since the setting is unmasked, there is no background to preserve, and thus standard metrics that rely on background consistency are not applicable. In addition to retaining the Whole metric (CLIP score between the entire edited image and the target prompt), we introduce ArcFace [6] for evaluating identity preservation, and CLIP-I for measuring similarity between the source and edited images. The results are reported in Table 2 below, which shows that our method achieves the best performance compared to others. Thanks to our proposed image inversion algorithm, which effectively learns the generative trajectory of the source image, our method can accomplish the intended edits.
>
> **Table 2: Quantitative results on facial images from FFHQ.**
> |Method|ArcFace↑|CLIP-I↑|Whole↑|
> |-|-|-|-|
> |NTI|0.56|0.83| 23.67 |
> |RF-Edit|0.61|0.79| 23.54  |
> |**Ours**|**0.63**|**0.86**|**24.82**|
>
> Note: Due to time constraints, we select representative methods with strong performance from both diffusion U-Net-based models and diffusion transformer-based models for comparison.
>
> [6] Jiankang Deng, et al. ArcFace: Additive Angular Margin Loss for Deep Face Recognition[C]//CVPR. 2019.
>
>
> **(b)  On-the-wild images.** Given that PIE-Bench contains nearly 50% natural images, it serves as a comprehensive benchmark for assessing our method. However, to specifically evaluate complex scenes, we select 20 MagicBrush images (due to time constraints) filtered by GPT-4o, comprising five samples each with **2, 3, 4, and 5 primary objects**. Table 3 demonstrates the superiority of our method in handling complex scenes.
>
> **Table 3: Quantitative results on the complex scene images from MagicBrush .**
> |Method|PSNR↑|LPIPS(×10³)↓|MSE(×10⁴)↓|SSIM(×10²)↑|Whole↑|Edited↑|
> |-|-|-|-|-|-|-|
> |NTI|8.81|452.03|1380.48|39.63|19.90|16.83|
> |RF-Edit|26.00|121.84|33.98|84.73|24.13|18.29|
> |**Ours**|**31.23**|**24.30**|**9.90**|**91.70**|**24.19**|**20.07**|
>
> > **Q3: What are the failure cases of the method in detail?**
>
> **A3:** We appreciate your valuable suggestion. In the final version, we will provide additional failure cases and elaborate on the limitation as follows.
>
> **Limitation.** While our method demonstrates strong performance across diverse editing tasks, it shows limitations in extreme cases such as style change, where no background needs to be preserved, and the image contains detailed structural patterns. In such cases, the blending between source and target tokens is constrained, which may lead to suboptimal preservation of structural fidelity from the original image. Nonetheless, thanks to our image inversion strategy, which effectively learns the generative trajectory of source image, our method can still accomplish intended edits, despite slight structural degradation. In contrast, other methods often fail in such challenging scenarios. For example, as shown in Figure 6, row 9 of the main paper, while other methods are unable to convert the painted bird into a realistic one, our method successfully achieves the style change, with only minor deviation in the bird's head pose.
>
> > **Q4: Is there are timing constraint on the method, as a two stage optimization is required for editing?**
>
> **A4:** We conduct a runtime comparison of our method and other methods on a single NVIDIA L20 GPU, measuring both inversion and editing time, as shown in Table 4 below. A key advantage lies in its efficiency during multiple edits on one image—a common real-world scenario. Once the inversion for a given image is completed, subsequent edits can be performed within **3.64 seconds**—over **7× faster** than other methods on average (initial inversion time only 4× longer than other methods on average). This design effectively front-loads the computational cost.
>
> **Table 4: Runtime comparison of inversion and editing.**
> |Method|Inversion|Once Editing|Twice Editing|
> |-|-|-|-|
> |P2P|14.40s|10.28s|10.28s|
> |MasaCtrl|**5.19s**|17.45s|17.45s|
> |P2P-Zero|13.31s|62.29s|62.29s|
> |NTI|95.54s|10.32s|10.32s|
> |PnP-Inv|8.32s|9.54s|9.54s|
> |NP|9.00s|10.37s|10.37s|
> |StableFlow|13.85s|27.20s|27.20s|
> |RF-Edit|55.48s|54.07s|54.07s|
> |**Ours**|107.06s|**3.64s**|**3.64s**|

---

> > ### Comment · Reviewer_LWWe · 2025-08-07
> > **Thank you for the rebuttal**
> >
> > Thanks to the authors for the rebuttal and the experiments reported during the limited time allocated for rebuttals. Overall my concerns regarding the sensitivity of the mask (with alternative extraction methods), facial attribute preservation during editing, timing constraints and on the wild images are addressed. I appreciate the authors reporting metrics such as ID preservation in this regard. Considering the reviews of the other reviewers and the fact that the paper makes a cntribution over a new type of generative model. I am increasing my score.

---

### Official Review · Reviewer_4PS6 · 2025-07-03

**Clarity:** 3
**Significance:** 2
**Originality:** 3
**Rating:** 4
**Confidence:** 2

**Summary:**

This paper proposes a method to adapt a type of pretrained text conditioned image generative models to the task of image editing. The model in question is Infinity, an autoregressive model on binary residual quantized tokens in the image scale space. The proposed method first conduct text inversion with learnable language style tokens and then performs editing with soft masking in the generation trajectory.

The proposed method is tested on PIE benchmark with add, delete, and change operations. It shows better image editing capability than other training-free baselines.

**Questions:**

I would like to encourage the authors provide more detailed explanation on the optimization of t_l and the blending of quantized tokens.

Also, I would like to see clarification on the experimental results. Are other baseline methods also using Inifinity as the base model? If not could the worse results for these methods be due to their worse base models?

**Ethical Concerns:**

["NO or VERY MINOR ethics concerns only"]

**Final Justification:**

The authors have addressed my concerns in the response. The method described with the added context seems reasonable and is achieving good editing quality. I am recommending accept.

**Limitations:**

Yes

**Quality:**

2

**Strengths And Weaknesses:**

Strengths:
+ The idea of using the Infinity model, which provides exact decomposition of images, as a trajectory to reference seems interesting. The proposed masking technique appears to provide good background preservation.
+ Based on the provided visual results, the proposed method can accurately achieve addition, deletion, and change.
+ The cost of adapting the pretrained text to image model is minimal thanks to the use of LoRA.

Weakness
- It is unclear how the text style token t_l is optimized or learned.
- The blending operation is not described clearly. Since the Infinity model uses quantized tokens, it is unclear how linear blending happens between two token indexes.
The method seems tailored to the infinity model; it is unclear whether it can be generalized to other types of image generative models.

---

> ### Author Rebuttal · Authors · 2025-07-30
>
> Thanks for your comments. We will explain your concerns point by point.
>
> **Main Comment**
>
> > **Q1: I would like to encourage the authors provide more detailed explanation on the optimization of t_l and the blending of quantized tokens.**
>
> **A1:** We appreciate the reviewer’s helpful suggestion. We now explain in detail (i) how our method optimizes $t_l$, and (ii) how the blending of quantized tokens is performed. We have also refined the narrative in the main paper to make these components clearer and better integrated.
>
> **(a) The optimization of $t_l$:**
>
> First, we introduce learnable prompt embeddings $t_l$, whose size is fixed to 20 tokens. Second, we pass both the source prompt $t_{sou}$ and the instruction prompt $t_{inv}$ (i.e., "the language style of this prompt is") through text encoder of Infinity $\Psi(\cdot)$ to obtain text embeddings $\Psi(t\_{sou}, t\_{ins})$. We then concatenate those embeddings with $t_l$ to form the textual conditioning input for Infinity. Finally, while keeping Infinity's weights frozen, we optimize only $t_l$ using cross-entropy loss, where the supervision signal comes from tokens $R_{1...K}^{sou}$ derived from the source image.
>
> **(b) The blending of quantized tokens:**
>
> (i) **The blending operation is performed on the quantized features $(\\{R_{k}\\}_{k=1}^K, R_k \in  \mathbb{R}^{h_k \times w_k \times d})$ rather than on indices**, which is formulated as $E_k^{tar} = \mathrm{Upsample}(R^{tar}_k, (h_K,w_K)) \odot (1-\mathcal{G}) + \mathrm{Upsample}(R^{sou}_k, (h_K,w_K)) \odot \mathcal{G}$.
>
> At each step $k$ of the autoregressive generation, the target token $R_k^{tar}$ is generated conditioned on the concatenation of the target prompt embedding and instruction embedding  $\Psi(t\_{tar},t\_{ins})$, and optimized $t_l$. It is then blended with source tokens $R_k^{sou}$ of the source image under the guidance of the piecewise linear smoothing kernel $\mathcal{G}$. Although this formulation is already provided in Algorithm 1 of the main paper, we appreciate the reviewer’s suggestion and will enhance the textual explanation in the final version to improve clarity and emphasis.
>
> (ii) The feasibility of performing linear blending at the quantized feature level is grounded in the fact that the Infinity decoder takes quantized features as input to generate images. As such, performing interpolation in the feature space is both reasonable and commonly adopted in generative modeling [1-3], as it supports smooth and semantically meaningful transitions.
>
> [1] Alec Radford, et al. Unsupervised Representation Learning with Deep Convolutional Generative Adversarial Networks[C]//ICLR. 2016.
>
> [2] Kingma D P, Welling M. Auto-encoding variational bayes[C]//ICLR. 2014.
>
> [3] Zhu J Y, et al. Toward multimodal image-to-image translation[C]//NeurIPS. 2017.
>
>
> > **Q2: Also, I would like to see clarification on the experimental results. Are other baseline methods also using Inifinity as the base model? If not could the worse results for these methods be due to their worse base models?**
>
> **A2:** This is an insightful question. Our method is a lightweight framework tailored to Infinity and indeed relies on the generative capacity of the base model. However, this dependency is not unique to our approach. We explicitly present base models used by all methods in Table 1 below and evaluate their generative performance on the GenEval benchmark [4], one of the most widely used benchmarks for assessing text-to-image generation quality. As shown in Table 2, **Infinity achieves comparable overall generative performance to the recently popular FLUX** [5], and even underperforms in certain tasks, such as two-object and counting. Nevertheless, **our Infinity-based image editing method significantly outperforms FLUX-based approaches** like StableFlow and RF-Edit, demonstrating the effectiveness of our method  spite the base model not having a clear advantage.
>
> **Table 1: Quantitative results on PIE-Bench.**
> |Method| Base Model | PSNR↑ | LPIPS(×10³)↓ | MSE(×10⁴)↓ | SSIM(×10²)↑ | Whole↑ | Edited↑ | IR(×10)↑ |
> |-|-|-|-|-|-|-|-|-|
> | P2P | Stable Diffusion v1.4 | 17.87| 208.80| 219.88| 71.14| 25.01| 22.44| 0.017|
> | MasaCtrl| Stable Diffusion v1.4 | 22.17| 106.62| 86.97| 79.67 | 23.96| 21.16| -1.66|
> | P2P-Zero| Stable Diffusion v1.4 | 20.44 | 172.22| 144.12| 74.67 | 22.80| 20.54| -6.59|
> | NTI| Stable Diffusion v1.4 | 27.03| 60.67| 35.86| 84.11   | 24.75 | 21.86| 2.77|
> | PnP-Inv| Stable Diffusion v1.5 | 22.46| 106.06 | 80.45  | 79.68 | 25.41| 22.62| 4.17|
> | NP| Stable Diffusion v1.4 | 26.21| 69.01| 39.73 | 83.40 | 24.61| 21.87 | 2.42|
> | StableFlow| FLUX.1-dev | 21.64| 92.28  | 115.21| 84.94| 24.65| 21.70| 1.88|
> | RF-Edit| FLUX.1-dev | 23.22| 131.18| 75.00 | 81.44  | 25.22| 22.40| 5.18 |
> | **EditInfinity (Ours)** | Infinity | **27.95** | **33.08**| **24.27** | **92.12** | **26.41** | **23.47**| **5.88**  |
>
> **Table 2: Evaluation on the GenEval benchmark.**
>
> |Base Model|Overall|Single Object |Two Object | Counting| Colors | Position | Attribute Binding |
> |-|-|-|-|-|-|-|-|
> | Stable Diffusion v1.4 |0.42 |0.97| 0.39|0.33|0.73|0.03|0.05|
> | Stable Diffusion v1.5 |0.43|0.97|0.38|0.35|0.76|0.04|0.06|
> |**FLUX.1-dev**|**0.66**|**0.98**|**0.81**|**0.74**|0.79|0.22|0.45|
> |**Infinity**|**0.66**|**0.98**|0.78|0.63|**0.83**|**0.25**|**0.53**|
>
> Note: When evaluating Infinity, we adopt the same evaluation protocol as used for Stable Diffusion v1.4, v1.5, and FLUX.1-dev, i.e., without prompt rewriting.
>
> [4] Dhruba Ghosh, et al. Geneval: An object-focused framework for evaluating text-to-image alignment[C]//NeurIPS. 2024.
>
> [5] Black Forest Labs. Flux. 2024.

---

> > ### Comment · Reviewer_4PS6 · 2025-08-05
> >
> > I would like to thank the authors for detailed responses. The reponse has addressed my question regarding the method description. The added table for editing results seems to be crucial. Please make sure to include it in the manuscript.
> >
> > For the prompt embedding, it is crucial to describe the prompt tuning approach as described in the response; otherwise, the audience may not understand what is happening.
> >
> > Given that my concerns are addressed, I am recommending borderline accept.

---

### Note · Authors · 2025-08-12

Dear Reviewer Ghm8,

We sincerely appreciate your insightful and constructive feedback. Your concerns regarding *Structured Editing Setup*, *Computation Cost*, and *Manual Design* were thoroughly explained and clarified during the rebuttal period. Below, we briefly summarize our explanation.

For *Structured Editing Setup*, we emphasized that our method is designed to operate with known editing regions rather than focusing on prompt understanding. To further validate its applicability in complex scenarios, we conducted additional experiments on open-ended editing tasks using 20 images from MagicBrush, with five samples each containing 2, 3, 4, and 5 primary objects. The results demonstrate that our method maintains strong performance even in challenging multi-object scenes.

For *Computation Cost*, we provided a runtime comparison with existing methods, measuring both inversion and editing times. While the initial inversion for a given image is only ~4 times slower than the average of other methods, subsequent edits can be performed within 3.64 seconds—over 7 times faster than competing methods on average. This demonstrates the significant advantage of our method for multiple edits on the same image, a common and practical application scenario.

For *Manual Design*, we explained that although our default setting uses user-provided masks, our method still achieves competitive performance when masks are automatically generated from cross-attention maps without modifying the framework. This demonstrates that our approach is not highly sensitive to the source of the mask.

Although we unfortunately did not receive confirmation during the rebuttal period on whether our responses fully resolved your concerns, we greatly appreciate the opportunity to further clarify our method. Your constructive suggestions have been highly valuable, and we have incorporated the corresponding revisions into the final version of the paper.

---

### Decision · Program_Chairs · 2025-09-17

**Decision:**

Accept (poster)

**Comment:**

This paper introduces EditInfinity, which is a text-based image editing method based on the autoregressive text-to-image generation model Infinity, which succeeds in high quality generation among vector quantized models available. The authors motivate the paper from the existing inversion-then-editing approach, where instead of inverting into a noisy latent (like in diffusion models), they initially invert the input text prompt by an extended embedding representation, which is followed by LoRA based adaptation for the reconstruction of precise details. Given a target text prompt, an input image and a region of interest specified by a mask, EditInfinity performs editing by manipulating the multi-scale quantized feature maps with a smoothed mask, then reconstruct the image using these manipulated feature maps. Authors demonstrate the effectiveness of the approach by using PIE-Bench, where they compare their approach with diffusion/flow-matching based editing methods. The provided quantitative analyses and qualitative results shows the effectiveness of the method.

The reviewers found the work somewhat incremental (the paper extends the Infinity model to editing tasks) but considered that there are some benefits to autoregressive image generation (such as speed) that merit further investigation in this area. Three of the reviewers were initially negative. They raised many concerns, which the authors addressed successfully. After discussion there were still some concerns about the method reliance on user-provided masks, a limitation noted by multiple reviewers. This requirement reduces its generalizability. However, since VQ-based approaches remain relatively underexplored, the reviewers believed the paper is an overall positive contribution. In the end, the reviewers were mildly positive towards publication of the paper.